# HiLo: A Learning Framework for Generalized Category Discovery Robust to Domain Shifts

**Hongjun Wang**[1]       **Sagar Vaze**[2]       **Kai Han**[1*]
[1]Visual AI Lab, The University of Hong Kong
[2]Visual Geometry Group, University of Oxford
hjwang@connect.hku.hk       sagar@robots.ox.ac.uk       kaihanx@hku.hk

## Abstract

Generalized Category Discovery (GCD) is a challenging task in which, given a partially labelled dataset, models must categorize all unlabelled instances, regardless of whether they come from labelled categories or from new ones. In this paper, we challenge a remaining assumption in this task: that all images share the same underline{domain}. Specifically, we introduce a new task and method to handle GCD when the unlabelled data also contains images from different domains to the labelled set. Our proposed 'HiLo' networks extract High-level semantic and Low-level domain features, before minimizing the mutual information between the representations. Our intuition is that the clusterings based on domain information and semantic information should be independent. We further extend our method with a specialized domain augmentation tailored for the GCD task, as well as a curriculum learning approach. Finally, we construct a benchmark from corrupted fine-grained datasets as well as a large-scale evaluation on DomainNet with real-world domain shifts, reimplementing a number of GCD baselines in this setting. We demonstrate that HiLo outperforms SoTA category discovery models by a large margin on all evaluations. Project page: https://visual-ai.github.io/hilo/

## 1 Introduction

*Category discovery* Han et al. (2019) has recently gained substantial interest in the computer vision community Han et al. (2020; 2021); Fini et al. (2021); Wen et al. (2023); Jia et al. (2021); Zhao & Han (2021). The task is to leverage knowledge from a number of labelled images, in order to discover and cluster images from novel classes in unlabelled data. Such a task naturally occurs in many practical settings; from products in a supermarket, to animals in the wild, to street objects for an autonomous vehicle. Specifically, Generalized Category Discovery (GCD) Vaze et al. (2022) has recently emerged as a challenging variant of the problem in which the unlabelled data can contain both instances from 'seen' and 'unseen' classes. As such, the problem is succinctly phrased as: *"given a dataset, some of which is labelled, categorise all unlabelled instances (whether or not they come from labelled classes)"*.

In this paper, we challenge a key, but often ignored, assumption in this setting: GCD methods still assume that all instances in the unlabelled set come from the same *domain* as the labelled data. In practise, unlabelled images may not only contain novel categories, but also exhibit low-level covariate shift Sun et al. (2022); Yan et al. (2019). It has long been established that the performance of image classifiers degrades substantially in the presence of such shifts Ganin et al. (2016); Tzeng et al. (2014); Zhang et al. (2019) and, indeed, we find that existing GCD models perform poorly in such a setting. Compared to related literature in, for instance, domain adaptation Du et al. (2021); Chen et al. (2022b); Zhu et al. (2023) or domain generalization Shi et al. (2022); Harary et al. (2022) the task proposed here presents a dual challenge: models must be *robust* to the low-level covariate shift while remaining *sensitive* to semantic novelty.

Concretely, we tackle a task in which a model is given access to labelled data from a source domain. It is further given access to a pool of unlabelled data, in which images may come from either the *source domain or new domains*, and whose categories may come from the *labelled classes or from new ones* (see Figure 1). Such a setting may commonly occur if, for example, images are taken with different cameras or under different weather conditions. Moreover, such a setting is often observed

---

*Corresponding author.

on the web, in which images come from many different domains and with innumerable concepts. We suggest that the ability to cluster novel concepts while accounting for such covariate shift will be an important factor in fully leveraging web-scale data.

To tackle these problems, we introduce the 'HiLo' architecture and learning framework. The HiLo architecture extracts both 'low-level' (early layer) and 'high-level' (late layer) features from a vision transformer Dosovitskiy et al. (2020). While extracting features at multiple stages of the network has been performed in domain adaptation Bousmalis et al. (2016); Peng et al. (2019b); Liu et al. (2020), we further introduce an explicit loss term to minimise mutual information between the two sets of features (Section 3.2.1).

The intuition is that the *covariate* and *semantic* information in the data is (by definition) independent, and that the inductive bias of deep architectures is likely to represent low-level covariate information in early layers, and abstract semantic information in later ones Olah et al. (2017); Zhou et al. (2021). Next, we take inspiration from a strong method from the domain adaptation field, Patch-Mix Zhu et al. (2023), which works by performing mixup augmentation in the embedding space of a pretrained transformer. While naive application of this method does not account for semantic novelty in unlabelled data, we extend the PatchMix objective to allow training with both a self-supervised contrastive objective (Section 3.2.2), and a semantic clustering loss (Section 3.2.2). With these changes, the PatchMix style augmentation is tailored to leverage both the labelled and unlabelled data available in the GCD setting. Our 'HiLo' feature design in our framework enables the model to disentangle domain and se-

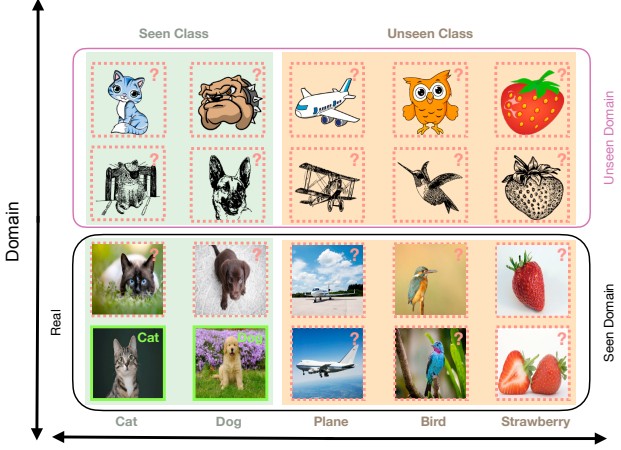

Figure 1: We present a new task where a model must categorize unlabelled instances from both seen and unseen categories, as well as seen and novel domains. In the example above, models are given labels only for the images in green boxes. The models are tasked with categorizing all unlabelled images, including those from different domains (top two rows) and novel categories (rightmost three columns on an orange background).

mantic features, while patch mixing allows the model to bridge the domain gap among images and focus more on determining the semantic shifts. Therefore, we introduce the patch mixing idea into our 'HiLo' framework, equipping it with a strong capability to discover novel categories from unlabelled images in the presence of domain shifts.

Finally, we find that *curriculum learning* Bengio et al. (2015); Zhou et al. (2020); Wu & Vorobeychik (2022) is particularly applicable to the setting introduced in this work (Section 3.2.3). Specifically, the quality of the learning signal differs substantially across different partitions of the data: from a clean supervised signal on the labelled set; to unsupervised signals from unlabelled data which *may or may not* come from the same domain and categories. It is non-trivial to train a GCD model to discover novel categories in the presence of both domain shifts and semantic shifts in the unlabelled data. To address this challenge, we introduce a curriculum learning approach which gradually increases the sampling probability weight of samples predicted as from unknown domains, as training proceeds. Our sequential learning process prioritizes the discovery of semantic categories initially and progressively enhances the model's ability to handle covariate shifts, which cannot be achieved by simply adopting existing domain adaptation methods.

To evaluate out models, we construct the 'SSB-C' benchmark suite – based on the recent Semantic Shift Benchmark (SSB) Vaze et al. (2021) – with domain shifts introduced by synthetic corruptions following ImageNet-C Hendrycks & Dietterich (2019). On this benchmark, as well as on a large-scale DomainNet evaluation with real data Peng et al. (2019a), we also reimplement a range of performant baselines from the category discovery literature. We find that, on both benchmarks, our method substantially outperforms all existing category discovery models Vaze et al. (2022); Wen et al. (2023); Han et al. (2019); Fini et al. (2021).

In summary, we make the following key contributions: **(i)** We formalize a challenging open-world task for category discovery in the presence of domain shifts; **(ii)** We develop a new method, HiLo, which disentangles covariate and semantic features to tackle the problem, extending state-of-the-art

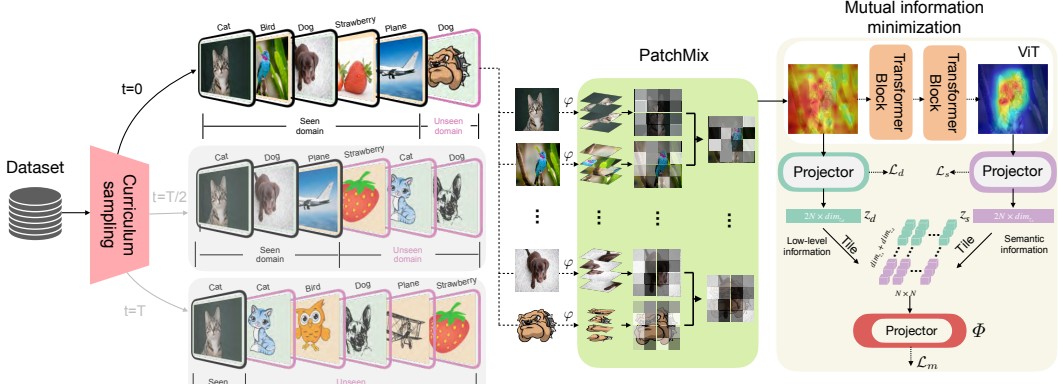

Figure 2: Overview of HiLo framework. Samples are drawn through our proposed curriculum sampling approach, considering the difficulty of each sample. Labelled and unlabelled samples are paired and augmented through PatchMix which we subtly adapt in the embedding space for contrastive learning for GCD. The mixed-up embeddings are then processed by our network with a high-level (for semantic) and low-level (for domain) feature design, allowing for the domain-semantic disentangled feature learning via mutual information minimization.

methods from the domain adaptation literature; **(iii)** We reimplement a range of category discovery models on a benchmark suite containing both fine-grained and coarse-grained datasets, with real and synthetic corruptions. **(iv)** We demonstrate that, on all datasets, our method substantially outperforms current state-of-the-art category discovery methods with finetuned hyperparameters.

## 2 RELATED WORK

**Category discovery** was firstly studied as novel category discovery (NCD) Han et al. (2019) and recently extended to generalized category discovery (GCD) Vaze et al. (2022). GCD extends NCD by including unlabelled images from both labelled and novel categories. Many successful NCD methods have been proposed (*e.g.*, DTC Han et al. (2019), RankStats Han et al. (2020; 2021), WTA Jia et al. (2021), DualRank Zhao & Han (2021), OpenMix Zhong et al. (2021b), NCL Zhong et al. (2021a), UNO Fini et al. (2021), class-relation knowledge distillation Gu et al. (2023)), they do not address domain shifts. Recent work Zang et al. (2023) considers domain shifts in NCD with labelled target domain images. We focus on GCD without any labelled instances from new domains, where unlabelled images may come from multiple novel domains. There is a growing body of literature for category discovery, including non-parametric methods (*e.g.*, GCD Vaze et al. (2022), CiPR Hao et al. (2023)); parametric methods (*e.g.*, SimGCD Wen et al. (2023)), ORCA Cao et al. (2021), $\mu$-GCD Vaze et al. (2023); prompt based techniques (*e.g.*, PromptCAL Zhang et al. (2023), SPTNet Wang et al. (2024a), PromptGCD Cendra et al. (2024)), debiased learning (*e.g.*, DebGCD Liu & Han (2025)), and other specialized methods (*e.g.*, GPC Zhao et al. (2023), DCCL Pu et al. (2023), InfoSieve Rastegar et al. (2024)). However, existing GCD methods do not consider domain shifts in unlabelled data.

**Semi-supervised learning** (SSL) aims to develop robust classification models using both labelled and unlabelled data, assuming instances belong to the same class set. Consistency-based approaches, such as Mean-teacher Tarvainen & Valpola (2017), Mixmatch Berthelot et al. (2019), and Fixmatch Sohn et al. (2020), have demonstrated effectiveness in SSL. Recent methods Chen et al. (2020b;c; 2021) have enhanced SSL performance by incorporating contrastive learning (e.g., Chen et al. (2020a), He et al. (2020)). Several studies Wang et al. (2022); Rizve et al. (2022); Wang et al. (2024b); Sun et al. (2024) have extended standard SSL to open-world settings.

**Unsupervised domain adaptation** (UDA) adapts models from a source domain to a target domain, with labelled data from the former and unlabelled data from the latter. UDA methods are categorized into moment matching Tzeng et al. (2014); Long et al. (2015; 2017); Zhang et al. (2019) and adversarial learning Ganin et al. (2016); Gao et al. (2021); Tang & Jia (2020) methods. DANN, FGDA, DADA are popular examples using a min-max game. MCD and SWD implicitly use adversarial learning with $L_1$ distance and sliced Wasserstein discrepancy, respectively. CGDM Du et al. (2021) leverages cross-domain gradient discrepancy, while Chen et al. (2022b) couples NWD with a single task-specific classifier with implicit K-Lipschitz constraint. PMTrans Zhu et al. (2023) aligns the source and target domains with the intermediate domain by employing semi-supervised mixup losses in both feature and label spaces. MCC Jin et al. (2020) minimizes between-class

confusion and maximizing within-class confusion, while NWD Chen et al. (2022b) uses a single task-specific classifier with implicit K-Lipschitz constraint to obtain better robustness for all the domain adaptation scenarios.

# 3 HiLo NETWORKS FOR GCD WITH DOMAIN SHIFTS

In this section, we start with the problem statement of GCD with domain shifts. Subsequently, we introduce the SimGCD baseline in Section 3.1, which serves as a robust GCD baseline upon which our method is built. Finally, we introduce our HiLo networks for GCD with domain shifts in Section 3.2.

**Problem statement.** We define *Generalized Category Discovery with domain shifts* as the task of classifying images from mixed domains $\Omega = \Omega^a \cup \Omega^b$ (where $\Omega^a \cap \Omega^b = \emptyset$ and $\Omega^b$ may contain multiple domains in practise), only having access to partially labelled samples from domain $\Omega^a$. The goal is to assign class labels to the remaining images, whose categories and domains may be seen or unseen in the labelled images. Formally, let $\mathcal{D}$ be an open-world dataset consisting of a labelled set $\mathcal{D}^l = \{(\boldsymbol{x}_i, y_i)\}_{i=1}^{N_l} \subset \mathcal{X}^l \times \mathcal{Y}^l$ and an unlabelled set $\mathcal{D}^u = \{\boldsymbol{x}_i\}_{i=1}^{N_u} \subset \mathcal{X}^u$. The label space for labelled samples is $\mathcal{Y}^l = \mathcal{C}_1$ and for unlabelled samples is $\mathcal{Y}^u = \mathcal{C} = \mathcal{C}_1 \cup \mathcal{C}_2$, where $\mathcal{C}, \mathcal{C}_1$, and $\mathcal{C}_2$ represent the label sets for 'All', 'Old', and 'New' categories, respectively. It is important to note that $\mathcal{Y}^l \subset \mathcal{Y}^u$. The objective of GCD with domain shifts is to classify all unlabelled images in $\mathcal{D}^u$ (from either $\Omega^a$ or $\Omega^b$) using only the labels in $\mathcal{D}^l$. This is different from the setting of NCD with domain shift Yu et al. (2022), which assumes $\mathcal{Y}^l \cap \mathcal{Y}^u = \emptyset$, and GCD, which assumes $\Omega^a = \Omega^b$ with $|\Omega^a| = |\Omega^b| = 1$. For notation simplicity, hereafter we omit the subscript $i$ for each image $\boldsymbol{x}_i$.

## 3.1 BACKGROUND: SimGCD

SimGCD Wen et al. (2023) is a representative end-to-end baseline for GCD, which integrates two primary losses for representation learning and parametric classification: (1) a contrastive loss $\mathcal{L}^{rep}$ based on InfoNCE Oord et al. (2018) is applied for the representation learning of the feature backbone; and (2) a cross-entropy loss $\mathcal{L}^{cls}$ for training a cosine classification head Gidaris & Komodakis (2018), utilizing different image views as pseudo-labels for one another. Following Vaze et al. (2022), SimGCD employs the ViT model as the backbone containing $m$ Transformer layers. Let $\mathcal{F}$ be the feature extractor consisting of these $m$ layers and $\mathcal{H}$ be a projection head. For an input image $\boldsymbol{x}$, a $\ell_2$-normalised feature can be obtained by $\boldsymbol{z} = \mathcal{H}(\mathcal{F}(\varphi(\boldsymbol{x})))$, where $\varphi$ is a standard embedding layer before the multi-head attention layers in the ViT model. The representation loss is

$$\mathcal{L}^{rep}(\boldsymbol{x}) = -\frac{1}{|\mathcal{P}(\boldsymbol{x})|} \sum_{\boldsymbol{z}^+ \in \mathcal{P}(\boldsymbol{x})} \log \sigma(\boldsymbol{z} \cdot \boldsymbol{z}^+; \tau), \tag{1}$$

where $\sigma(\cdot; \tau)$ is the softmax operation with a temperature $\tau$ for scaling and $\mathcal{P}(\boldsymbol{x})$ denotes the positive feature set for each $\boldsymbol{x}$. Suppose we sample a batch $\mathcal{B}$, which contains labelled images and unlabelled images, denoted as $\mathcal{B}^l$ and $\mathcal{B}^u$, respectively. For each $\boldsymbol{x} \in \mathcal{B}$ (either a labelled or unlabelled image), $\mathcal{P}(\boldsymbol{x})$ contains only the feature of a different view of the same image. For each $\boldsymbol{x} \in \mathcal{B}^l$, an additional $\mathcal{P}(\boldsymbol{x})$ including features of other images from the same class and the feature of a different view of the same image is also used for supervised contrastive learning. Likewise, the classification loss can be written as

$$\mathcal{L}^{cls}(\boldsymbol{x}) = -\sum_{\boldsymbol{w} \in \boldsymbol{W}} \boldsymbol{q} \log \sigma(\hat{\boldsymbol{z}} \cdot \boldsymbol{w}; \tau), \tag{2}$$

where $\boldsymbol{W}$ is a set of prototypes and each vector $\boldsymbol{w}$ in $\boldsymbol{W}$ represents a $\ell_2$-normalised learnable class prototype. $\hat{\boldsymbol{z}}$ is the $\ell_2$-normalised vector of $\mathcal{F}(\varphi(\boldsymbol{x}))$. For each $\boldsymbol{x} \in \mathcal{B}$, $\boldsymbol{q}$ is the pseudo-label from a sharpened prediction of a different view of the same image. For each $\boldsymbol{x} \in \mathcal{B}^l$, an additional $\boldsymbol{q}$ as the one-hot ground-truth vector is also used for supervised learning. Let $\mathcal{L}^{r,c}$ be the summation of $\mathcal{L}^{rep}$ and $\mathcal{L}^{cls}$ for simplification, the overall loss can then be written as:

$$\mathcal{L}_{sim} = \lambda \sum_{\boldsymbol{x} \in \mathcal{B}} \mathcal{L}^{r,c}(\boldsymbol{x}) + (1 - \lambda) \sum_{\boldsymbol{x} \in \mathcal{B}^l} \mathcal{L}^{r,c}(\boldsymbol{x}) + \epsilon \Delta, \tag{3}$$

where $\mathcal{B}^l$ denotes the subset of labelled samples in the current mini-batch, and $\Delta$ is an entropy maximization term to prevent pseudo-label collapse Assran et al. (2022). Finally, $\lambda$ and $\epsilon$ are hyperparameters, and we refer to the original work for further details Wen et al. (2023).

Despite achieving strong performance on the standard single-domain GCD task, SimGCD struggles in the more realistic scenario in which the unlabelled data exhibits *domain shifts*. Next, we present our HiLo framework, which builds upon SimGCD and introduces three key innovations to effectively handle domain shifts in GCD.

## 3.2 HiLo: High and Low-level Networks

The architecture of our HiLo framework is outlined in Figure 2. Firstly, we propose a method to disentangle domain features and semantic features using mutual information minimization. Secondly, we introduce patch-wise mixup augmentation in the image embeddings, facilitating knowledge transfer between labelled and unlabelled data across different domains. Lastly, we employ a curriculum sampling scheme that gradually increases the proportion of samples from the unseen domain during training. This curriculum-based approach aids the learning process by initially focusing on easier single-domain discrimination and gradually transitioning to more challenging cross-domain discrimination.

### 3.2.1 Learning domain-semantic disentangled features for GCD

As covariate shift observed by new domains $\Omega$ in $\mathcal{D}^u$ degrades performance, we aim to learn two distinct feature sets encoding domain and semantic aspects by minimizing their mutual information. For each image $\boldsymbol{x}$, we thus consider that its feature can be partitioned into two parts, depicting domain-specific (*e.g.*, *real*, *sketch*) and semantic information (*e.g.*, *cat*, *dog*), respectively. However, it is intractable to estimate the mutual information between random variables of semantic and domain in finite high-dimensional space without parametric assumptions Zhao et al. (2018); Song & Ermon (2019). Instead of calculating the exact value, assumptions based on convex conjugate Nguyen et al. (2010) and GAN Nowozin et al. (2016) are utilized for estimation. Belghazi et al. (2018); Hjelm et al. (2018) further demonstrate that this estimation can be achieved without such assumptions. We thus adopt the approach from Hjelm et al. (2018) based on Jensen-Shannon divergence to estimate the mutual information. For each image, instead of considering a single feature vector as $\boldsymbol{z} = \mathcal{H}(\mathcal{F}(\varphi(\boldsymbol{x})))$, here we consider two feature vectors, $\boldsymbol{z}_d$ and $\boldsymbol{z}_s$, for domain and semantic information respectively. Inspired by the fact that deeper layers of the model give higher-level features and the shallower layers of the model give lower-level features Sze et al. (2017); Zhou et al. (2021), we use the feature from the very first layer of the ViT as $\boldsymbol{z}_d$ and that from the very last layer as $\boldsymbol{z}_s$. Specifically, we obtain $[\boldsymbol{z}_d, \boldsymbol{z}_s] = \tilde{\mathcal{H}}(\mathcal{F}(\varphi(\boldsymbol{x})))$, where $\tilde{\mathcal{H}}$ consists of two projection heads, one on the first layer feature of $\mathcal{F}$ and the other on the last layer feature of $\mathcal{F}$ (see Figure 2). Therefore, the mutual information between domain and semantic features can be approximated by a Jensen-Shannon estimator:

$$\mathcal{L}_m = I_\Phi(\boldsymbol{z}_d, \boldsymbol{z}_s) = \mathbb{E}_{p(\boldsymbol{z}_d, \boldsymbol{z}_s)}\left[-\log\left(1 + e^{-\Phi(\boldsymbol{z}_d, \boldsymbol{z}_s)}\right)\right] - \mathbb{E}_{p(\boldsymbol{z}_d)p(\boldsymbol{z}_s)}\left[\log\left(1 + e^{\Phi(\boldsymbol{z}_d, \boldsymbol{z}_s)}\right)\right],$$
(4)

where $\Phi$ is an MLP and an output dimension of 1. $\Phi$ takes the concatenation of $\boldsymbol{z}_s$ and $\boldsymbol{z}_d$ as input and predict a single scalar value. We aim to minimize the expected log-ratio of the joint distribution concerning the product of marginals. Note that here $\boldsymbol{z}_s$ and $\boldsymbol{z}_d$ may come from two different images. In practice, we tile the domain and semantic features of all the images in the mini-batch, and concatenate them, before applying $\Phi$ on all the concatenated features. We then extract the diagonal entries (which are from the joint distribution) as the first term and the other entries (which are from the marginals) as the second term in Equation (4).

### 3.2.2 PatchMix contrastive learning

Mixup Zhang et al. (2018b) is a powerful data augmentation technique that involves blending pairs of samples and their corresponding labels to create new synthetic training examples. It has been shown to be very effective in semi-supervised learning Hataya & Nakayama (2019), long-tailed recognition Xu et al. (2021), etc. In the presence of domain shifts, Mixup has also been shown to be effective in unsupervised domain adaptation Na et al. (2021) and domain generalization Zhang et al. (2018a); Yun et al. (2019); Zhou et al. (2021). Recently, PMTrans Zhu et al. (2023) introduced PatchMix, which is a variant of Mixup augmentation by mixing up the embeddings of images in the Transformer-based architecture for domain adaptation. Particularly, for an input image $\boldsymbol{x}$ with label $y$, PatchMix augments its $j$-th embedding patch by

$$\bar{\varphi}(\boldsymbol{x})_j = \beta_j \odot \varphi(\boldsymbol{x})_j + (1 - \beta_j) \odot \varphi(\boldsymbol{x}')_j,$$
(5)

where $\boldsymbol{x}'$ is an unlabelled image with or without domain shift, $\beta_j \in [0, 1]$ is the random mixing proportion for the $j$-th patch, sampled from Beta distribution, and $\odot$ denotes the multiplication

operation. A one-hot vector derived from $y$ is then smoothed based on $\beta_j$ to supervise the cross-entropy loss to train the classification model. However, this works under the assumption that the out-of-domain samples share the same class space with the in-domain samples, restricting its application to the more practical scenarios where the out-of-domain samples may come from new classes as we consider in the problem of GCD with domain shift. Hence, we devise a PatchMix-based contrastive learning method to address the challenge of GCD in the presence of domain shift. Our approach properly leverages all available samples, including both labelled and unlabelled data, from both in-domain and out-of-domain sources, encompassing both old and new classes. By incorporating these diverse samples, our technique aims to improve the model's ability to handle domain shifts and effectively generalize across different classes.

When incorporating the PatchMix into our problem setting, the unlabelled sample $x'$ in Equation (5) may have both domain and semantic shifts. With the new PatchMix augmented embedding layer $\bar{\varphi}$ and the two projection heads of $\tilde{\mathcal{H}}$, we can obtain $[\bar{z}_d, \bar{z}_s] = \tilde{\mathcal{H}}(\mathcal{F}(\bar{\varphi}(x)))$. We separately consider the learning of domain and semantic features. For semantic features, we introduce a factor $\alpha$ which takes the portion semantic of the sample $x$ into account, after mixing up with $x'$. In specific, the contrastive loss in Equation (1) is now modified as:

$$\mathcal{L}_s^{rep}(x) = -\frac{1}{|\mathcal{P}(x)|} \sum_{\bar{z}_s^+ \in \mathcal{P}(x)} \alpha \log \sigma(\bar{z}_s \cdot \bar{z}_s^+; \tau), \tag{6}$$

where $\alpha = \frac{\beta \cdot s}{\beta \cdot s + (1-\beta) \cdot s'}$. $\beta$ denotes the vector consisting of all $\beta_j$ as in Equation (5). $s$ and $s'$ are two vectors storing the attention scores for all the patches for $x$ and $x'$ respectively. The attention scores, computed following Chen et al. (2022a); Zhu et al. (2023), account for the semantic weight of each patch. To train the semantic classification head, we adopt the loss as in Equation (2). Differently, if $x$ is a labelled sample, inspired by Szegedy et al. (2016), we replace $q$ with $\bar{q} = \alpha \cdot q + \frac{1-\alpha}{|\mathcal{C}|} \cdot \mathbf{1}$, where $q$ is the one-hot vector derived from the label $y$ of $x$. If $x$ is unlabelled, similar to Equation (2), $q$ is a pseudo-label from a sharpened prediction of another mixed-up view. Aside from the label $q$, we also need to learn another set of semantic prototypes by replacing $W$ with $W^s$. Let the modified classification loss be $\mathcal{L}_s^{cls}$ and $\mathcal{L}_s^{r,c}$ be the summation of $\mathcal{L}_s^{rep}$ and $\mathcal{L}_s^{cls}$. The loss for PatchMix-based semantic representation and classification learning is:

$$\mathcal{L}_s = \lambda \sum_{x \in \mathcal{B}} \mathcal{L}_s^{r,c}(x) + (1 - \lambda) \sum_{x \in \mathcal{B}^l} \mathcal{L}_s^{r,c}(x). \tag{7}$$

Next, for domain-specific features, we employ the same loss as Equation (6), except that we now train on $\bar{z}_d$, for representation learning. We denote this loss as $\mathcal{L}_d^{rep}$. For training the classification head, we again adopt the loss as in Equation (2) but modify the label $q$ and learn a set of domain prototypes $W^d$. Therefore, if $x$ is a labelled sample, $\bar{q} = \alpha \cdot q + \frac{1-\alpha}{|\Omega|} \cdot \mathbf{1}$, where $q$ is the domain label. Note that we only assume that the labelled samples are from the same domain and do not assume that domain labels are available for any unlabelled samples, which is more realistic and challenging. Therefore, the only known domain label is typically 1. To obtain pseudo-labels for the unlabelled samples, we run the semi-supervised $k$-means as in Vaze et al. (2022) on the current mini-batch. We denote this modified classification loss as $\mathcal{L}_d^{cls}$ and the summation of $\mathcal{L}_d^{rep}$ and $\mathcal{L}_d^{cls}$ as $\mathcal{L}_d^{r,c}$. Therefore, the loss for representation and classification learning can be written as

$$\mathcal{L}_d = \lambda \sum_{x \in \mathcal{B}} \mathcal{L}_d^{r,c}(x) + (1 - \lambda) \sum_{x \in \mathcal{B}^l} \mathcal{L}_d^{r,c}(x). \tag{8}$$

**Overall loss.** We apply the mean-entropy maximization regularizer, as described in Equation (3), to both semantic and domain feature learning. These regularizers are denoted as $\Delta_s$ and $\Delta_d$ respectively. Let $\Delta = \Delta_s + \Delta_d$ and $\varepsilon$ be the balance factor. The overall loss for our HiLo framework can then be written as

$$\mathcal{L} = \mathcal{L}_m + \mathcal{L}_s + \mathcal{L}_d + \varepsilon \Delta. \tag{9}$$

### 3.2.3 CURRICULUM SAMPLING

As curriculum sampling Bengio et al. (2015) can effectively enhance the generalization capability of models by gradually increasing the difficulty of the training data, which is also a natural fit to the GCD with domain shift problem. Here, we also introduce a curriculum sampling scheme to further

enhance the learning of our HiLo framework. We expect the training to start by focusing on samples from the same domain to learn semantic features and leverage more samples containing the additional challenge of domain shifts in the later training stages. To this end, we devise a difficulty measure $p_{cs}(\boldsymbol{x}|t)$ for each sample $\boldsymbol{x}$ at training time step $t$ (*i.e.*, epoch), by considering the portion of samples belonging to each domain. As the unlabelled samples are from multiple domains and we do not have access to the domain label, we run the semi-supervised $k$-means on all the domain features extracted using the DINO pretrained backbone. Let the resulting clusters along the domain axis be $\hat{\mathcal{D}}^a$ and $\hat{\mathcal{D}}^b$, which corresponds to domains $\Omega^a$ and $\Omega^b$ respectively and $\mathcal{D}^u = \hat{\mathcal{D}}^a \cup \hat{\mathcal{D}}^b$. With the above, we then define the sampling probability weight $p_{cs}(\boldsymbol{x}|t)$ for each sample as follows:

$$
p_{cs}(\boldsymbol{x}|t) = \begin{cases} 1, & \boldsymbol{x} \in \mathcal{D}^l \\ \dfrac{|\mathcal{D}^l|}{|\hat{\mathcal{D}}^a|}, & \boldsymbol{x} \in \hat{\mathcal{D}}^a, \\ r_0 + (r' - r_0)\mathbb{1}(t > t'), & \boldsymbol{x} \in \hat{\mathcal{D}}^b \end{cases} \tag{10}
$$

where $\mathbb{1}(\cdot)$ is an indicator function, $t'$ is a constant epoch number since which we would like to increase the portion of samples from unknown domains, $r_0$ and $r'$ are constant probabilities for samples from unknown domains to be sampled in the earlier stages (*i.e.*, $< t'$) and latter stages (*i.e.*, $> t'$), $t$ indicates the current training time step. In our formulation, (1) if $\boldsymbol{x}$ is a labelled sample, its $p_{cs}(\boldsymbol{x}|t)$ is set to 1, without any discount; (2) if $\boldsymbol{x}$ is an unlabelled sample and is in $\hat{\mathcal{D}}^a$ (*i.e.*, predicted as from the seen domain), $p_{cs}(\boldsymbol{x}|t)$ is set to $\frac{|\mathcal{D}^l|}{|\hat{\mathcal{D}}^a|}$ (*i.e.*, proportional to the labelled and unlabelled samples from *the same domain*, as per the sampling strategy used in the conventional GCD without domain shifts Vaze et al. (2022)); and (3) if $\boldsymbol{x}$ is an unlabelled sample and is in $\hat{\mathcal{D}}^b$ (*i.e.*, predicted as from the *unseen domain*), its $p_{cs}(\boldsymbol{x}|t)$ will increase along with the training after epoch $t'$. We investigate choices of $r_0$, $r'$ and $t'$ in Appendix M.

In Appendix D, we provide an approximated theoretical analysis for our method with two theorems. Theorem 1 suggests (1) that learning on the original domain data first can effectively lower the error bound of category discovery on $\mathcal{D}^u$ and (2) the domain head that can reliably discriminate original and new domain samples can further reduce this error bound. Theorem 2 suggests that minimizing the mutual information between domain and semantic features can further lower the error bound of category discovery on $\mathcal{D}^u$. These theorems further validate the effectiveness of our method from a theoretical perspective.

## 4 EXPERIMENTS

### 4.1 EXPERIMENTAL SETUP

**Datasets.** To validate the effectiveness of our method, we perform various experiments on the largest public datasets with domain shifts, DomainNet Peng et al. (2019a), containing about 0.6 million images with 345 categories distributed among six domains. Moreover, based on the Semantic Shift Benchmark (SSB) Vaze et al. (2021) (including CUB Welinder et al. (2010), Stanford Cars Krause et al. (2013b), and FGVC-Aircraft Maji et al. (2013)), we construct a new corrupted dataset called SSB-C (*i.e.*, CUB-C, Scars-C, and FGVC-C) following Hendrycks & Dietterich (2019). We exclude unrealistic corruptions and corruptions that may lead to domain leakage to ensure that the model does not see any of the domains in SSB-C during training (see Appendix A for details). Overall, we introduce 9 types of corruption and 5 levels of corruption severity for each type, resulting in a dataset $45\times$ larger than SSB. For the semantics axis, on both DomainNet and SSB-C, following Vaze

Table 1: Statistics of the evaluation datasets.

| Dataset | Labelled | | | Unlabelled | | |
|---|---|---|---|---|---|---|
| | #Image | #Class $|\mathcal{Y}^l|$ | #Domain $|\Omega^a|$ | #Image | #Class $|\mathcal{Y}^u|$ | #Domain $|\Omega|$ |
| DomainNet | 39.1K | 172 | 1 | 547.5K | 345 | 6 |
| CUB-C | 1.5K | 100 | 1 | 45K | 200 | 10 |
| Scars-C | 2.0K | 98 | 1 | 61K | 196 | 10 |
| FGVC-C | 1.7K | 50 | 1 | 50K | 100 | 10 |

et al. (2022), we sample a subset of all classes as the old classes and use 50% of the images from these labelled classes to construct $\mathcal{D}^l_{\Omega^a}$. The remaining images with both old classes and new classes

are treated as the unlabelled data $\mathcal{D}_{\Omega^a}^u$. For the domain axis, on DomainNet, we select images from the 'real' domain as $\mathcal{D}_{\Omega^a}$ and pick one of the remaining domains as $\mathcal{D}_{\Omega^b}$ in turn (or include all the remaining domains as $\mathcal{D}_{\Omega^b}$). While on SSB-C, we use each dataset in SSB as $\mathcal{D}_{\Omega^a}$ and its corresponding corrupted dataset in SSB-C as $\mathcal{D}_{\Omega^b}$. Statistics of the datasets are shown in Table 1.

**Evaluation protocol.** For DomainNet, $\Omega^a = \{\omega_1\}$ and $\Omega^b = \{\omega_2\}$, where $\omega_i$ stands for different domains. We also experiment with the case where $\Omega^b = \{\omega_2, \cdots, \omega_6\}$. We train the models on $\mathcal{D}_{\Omega^a}$ (*i.e.*, $\mathcal{D}_{\Omega^a}^l \cup \mathcal{D}_{\Omega^a}^u$) and $\mathcal{D}_{\Omega^b}^u$ of all classes without annotations. For SSB-C, $\Omega^a = \{\omega_1\}$ and $\Omega^b = \{\omega_2, \cdots, \omega_{10}\}$ since we have nine types of corruptions. During evaluation, we compare the ground-truth labels $y_i$ with the predicted labels $\hat{y}_i$ and measure the clustering accuracy as $ACC = \frac{1}{|D^u|} \sum_{i=1}^{|D^u|} \mathbb{1}(y_i = \phi(\hat{y}_i))$, where $\phi$ is the optimal permutation that matches the predicted cluster assignments to the ground-truth labels. We report the ACC values for 'All' classes (*i.e.*, instances from $\mathcal{Y}$), the 'Old' classes subset (*i.e.*, instances from $\mathcal{Y}^l$), and 'New' classes subset (*i.e.*, instances from $\mathcal{Y}^u$) for $\mathcal{D}_{\Omega^a}^u$ and $\mathcal{D}_{\Omega^b}^u$ separately.

**Implementation details.** Following the common practice in GCD, we use the DINO Caron et al. (2021) pre-trained ViT-B/16 as the feature backbone and the number of categories is known as in Wen et al. (2023) for all methods for fair comparison. When the category number is unknown, one can employ existing methods (*e.g.*, Han et al. (2019); Vaze et al. (2022); Hao et al. (2023); Zhao et al. (2023)) to estimate it and substitute it into the category discovery methods (see Table 24). The 768-dimensional embedding vector corresponding to the CLS token is used as the image feature. For the feature backbone, we only fine-tune the last Transformer layer. We train each dataset for $T = 200$ epochs using a batch size of 256. We follow the protocol in Vaze et al. (2022); Wen et al. (2023) to select the optimal hyperparameters for our method and all baselines, based on the 'All' accuracy on the validation split of $\mathcal{D}_{\omega_1}$. The initial learning rate for our approach is 0.1 for CUB and 0.05 for other datasets, and the rate is decayed using a cosine schedule. $t'$ is set to the 80-th epoch. $r_0$ is assigned as $|\mathcal{D}^l|/|\hat{\Omega}^b|$ for DomainNet and 0 for SSB-C. $r'$ is set to 1 for DomainNet and 0.05 for SSB-C. $\epsilon$ is set to 0.1. Following Vaze et al. (2022); Wen et al. (2023), we set $\lambda = 0.35$. See Appendices M and N for choices of hyperparameters for HiLo components and learning rates for all methods.

## 4.2 MAIN COMPARISON

We compare our method with ORCA Cao et al. (2021), GCD Vaze et al. (2022) and SimGCD Wen et al. (2023) in generalized category discovery, along with two strong baselines RankStats+ Han et al. (2021) and UNO+ Fini et al. (2021) adapted from novel category discovery, on DomainNet (Table 2) and SSB-C (Table 3), respectively. Additionally, we provide results by incorporating UDA techniques in Section 4.3 and the strong CLIP model in Appendix G.

In Table 2, we present results on DomainNet considering one domain as $\Omega^b$ each time. Our method consistently outperforms other methods for 'All' classes (even better for 'New' classes) in both domain $\Omega^a$ and $\Omega^b$ by a large margin. For example, for the 'Real' and 'Painting' pair, it outperforms the GCD SoTA method, SimGCD, by nearly 5% and 19% in proportional terms, which is remarkable considering the gap between different methods. RankStat+ performs well on 'Old' categories in the unseen domain $\Omega^b$. In Appendix B, we present results on DomainNet considering all domains except 'Real' as $\Omega^b$. It can be seen, in such a challenging mixed domain scenario, our method still substantially outperforms other methods. A breakdown evaluation of each domain shift for both datasets can be found in Appendices H and I.

Table 2: Evaluation on the DomainNet dataset. The model is trained on the 'Real' (*i.e.*, $\Omega^a$) + 'Painting'/'Sketch'/'Quickdraw'/'Clipart'/'Infograph' (*i.e.*, $\Omega^b$) domains in turn.

| | Real+Painting | | | | | | Real+Sketch | | | | | | Real+Quickdraw | | | | | | Real+Clipart | | | | | | Real+Infograph | | | | | |
|---|---|---|---|---|---|---|---|---|---|---|---|---|---|---|---|---|---|---|---|---|---|---|---|---|---|---|---|---|---|---|
| | Real | | | Painting | | | Real | | | Sketch | | | Real | | | Quickdraw | | | Real | | | Clipart | | | Real | | | Infograph | | |
| Methods | All | Old | New | All | Old | New | All | Old | New | All | Old | New | All | Old | New | All | Old | New | All | Old | New | All | Old | New | All | Old | New | All | Old | New |
| RankStats+ | 34.1 | 62.0 | 19.7 | 29.7 | **49.7** | 9.6 | 34.2 | 62.0 | 19.8 | 17.1 | **31.1** | 6.8 | 34.1 | 62.5 | 19.5 | 4.1 | 4.4 | 3.9 | 34.0 | 62.4 | 19.4 | 24.1 | **45.1** | 6.2 | 34.2 | 62.4 | 19.6 | 12.5 | **21.9** | 6.3 |
| UNO+ | 44.2 | 72.2 | 29.7 | 30.1 | 45.1 | 17.2 | 43.7 | 72.5 | 28.9 | 12.5 | 17.0 | 9.2 | 31.1 | 60.0 | 16.1 | 6.3 | 5.8 | 6.8 | 44.5 | 66.1 | 33.3 | 21.9 | 35.6 | 10.1 | 42.8 | 69.4 | 29.0 | 10.9 | 15.2 | 8.0 |
| ORCA | 31.9 | 49.8 | 23.5 | 28.7 | 38.5 | 7.1 | 32.5 | 50.0 | 23.9 | 11.4 | 14.5 | 7.2 | 19.2 | 39.1 | 15.3 | 3.4 | 3.5 | 3.2 | 32.0 | 49.7 | 23.9 | 19.1 | 31.8 | 4.3 | 29.1 | 47.7 | 20.1 | 8.6 | 13.7 | 7.1 |
| GCD | 47.3 | 53.6 | 44.1 | 32.9 | 41.8 | 23.0 | 48.0 | 53.8 | 45.3 | 16.6 | 22.4 | 11.1 | 37.6 | 41.0 | 35.2 | 5.7 | 4.2 | 6.9 | 47.7 | 53.8 | 44.3 | 22.4 | 34.4 | 16.0 | 41.9 | 46.1 | 39.0 | 10.9 | 17.1 | 8.8 |
| SimGCD | 61.3 | **77.8** | 52.9 | 34.5 | 35.6 | 33.5 | 62.4 | 77.6 | 54.6 | 16.4 | 20.2 | 13.6 | 47.4 | 64.5 | 37.4 | 6.6 | 5.8 | 7.5 | 61.6 | 77.2 | 53.6 | 23.9 | 31.5 | 17.3 | 52.7 | 67.0 | 44.8 | 11.6 | 15.4 | 9.1 |
| HiLo (Ours) | **64.4** | 77.6 | **57.5** | **42.1** | 42.9 | **41.3** | **63.3** | **77.9** | **55.9** | **19.4** | 22.4 | **17.1** | **58.6** | **76.4** | **52.5** | **7.4** | **6.9** | **8.0** | **63.8** | **77.6** | **56.6** | **27.7** | 34.6 | **21.7** | **64.2** | **78.1** | **57.0** | **13.7** | 16.4 | **11.9** |

In Table 3, we show the results on SSB-C. We can see that HiLo significantly outperforms other methods across the board. For example, on CUB-C, HiLo outperforms SimGCD nearly 43.8% in proportional terms within $\Omega^a$ and 51.4% on unlabelled samples within $\Omega^b$. SimGCD shows good performance for new categories, while UNO+ demonstrates good performance for old categories.

Comparing the results with the single domain results in Vaze et al. (2022); Wen et al. (2023), we find that including corrupted data during training impairs the performance on the original domain. SSB-C

Table 3: Evaluation on SSB-C datasets. We report results of baselines in the seen domain (*i.e.*, Original) and the overall performance of different corruptions (*i.e.*, Corrupted). On 'Corrupted', our model provides between **20% and 80% relative gains** over SimGCD Wen et al. (2023).

| | CUB-C | | | | | | Scars-C | | | | | | FGVC-C | | | | | |
|---|---|---|---|---|---|---|---|---|---|---|---|---|---|---|---|---|---|---|
| | Original | | | Corrupted | | | Original | | | Corrupted | | | Original | | | Corrupted | | |
| Methods | All | Old | New | All | Old | New | All | Old | New | All | Old | New | All | Old | New | All | Old | New |
| RankStats+ | 19.3 | 22.0 | 15.4 | 13.6 | 23.9 | 4.5 | 14.8 | 20.8 | 7.8 | 11.5 | 22.6 | 1.0 | 14.4 | 16.4 | 14.5 | 8.3 | 15.6 | 5.0 |
| UNO+ | 25.9 | 40.1 | 21.3 | 21.5 | 33.4 | 8.6 | 22.0 | 41.8 | 7.0 | 16.9 | 29.8 | 4.5 | 22.0 | 33.4 | 15.8 | 16.5 | 25.2 | 8.8 |
| ORCA | 18.2 | 22.8 | 14.5 | 21.5 | 23.1 | 18.9 | 19.1 | 28.7 | 11.2 | 15.0 | 22.4 | 8.3 | 17.6 | 19.3 | 16.1 | 13.9 | 17.3 | 10.1 |
| GCD | 26.6 | 27.5 | 25.7 | 25.1 | 28.7 | 22.0 | 22.1 | 35.2 | 20.5 | 21.6 | 29.2 | 10.5 | 25.2 | 28.7 | 23.0 | 21.0 | 23.1 | 17.3 |
| SimGCD | 31.9 | 33.9 | 29.0 | 28.8 | 31.6 | 25.0 | 26.7 | 39.6 | 25.6 | 22.1 | 30.5 | 14.1 | 26.1 | 28.9 | 25.1 | 22.3 | 23.2 | 21.4 |
| UniOT | 27.5 | 29.3 | 26.8 | 27.3 | 33.2 | 22.5 | 24.3 | 37.5 | 22.3 | 22.9 | 31.4 | 13.7 | 27.3 | 29.8 | 22.5 | 21.6 | 23.5 | 19.6 |
| HiLo (Ours) | 56.8 | 54.0 | 60.3 | 52.0 | 53.6 | 50.5 | 39.5 | 44.8 | 37.0 | 35.6 | 42.9 | 28.4 | 44.2 | 50.6 | 47.4 | 31.2 | 29.0 | 33.4 |

is $45\times$ larger than SSB, posing a significant challenge and resulting in unsatisfactory performance for existing methods. However, our method, HiLo, continues to demonstrate promising results, further validating its effectiveness.

Table 4: Influence of different model components. We select the 'Real' and 'Painting' domains from DomainNet to train the DINO model with the techniques introduced above as the baseline. Rows 2-4 indicate our main conceptual methodological contributions and rows 5-7 represent the careful ablation of engineering choices.

| | | Real | | | Painting | | |
|---|---|---|---|---|---|---|---|
| | Methods | All | Old | New | All | Old | New |
| Reference | SimGCD Wen et al. (2023) | 61.3 | 77.8 | 52.9 | 34.5 | 35.6 | 33.5 |
| (1) | SimGCD + PatchMix in Zhu et al. (2023) | 62.5 | 76.3 | 54.2 | 34.8 | 36.0 | 33.8 |
| (2) | SimGCD + PatchMix for CL | 63.5 | 75.0 | 57.6 | 36.6 | 39.6 | 33.6 |
| (3) | SimGCD + Disentangled Features | 66.4 | 79.2 | 59.8 | 35.6 | 36.7 | 34.2 |
| (4) | SimGCD + Curriculum Sampling | 63.6 | 78.6 | 55.9 | 38.4 | 39.9 | 35.9 |
| Reference | HiLo | 64.4 | 77.6 | 57.5 | 42.1 | 42.9 | 41.3 |
| (5) | $z_d, z_s$ from deep features only | 28.2 | 40.3 | 22.7 | 13.6 | 20.0 | 11.0 |
| (6) | $z_d, z_s$ from shallow features only | 10.1 | 18.1 | 6.4 | 5.7 | 9.2 | 5.7 |
| (7) | Self-dist. for domain head | 63.2 | 76.8 | 56.1 | 40.2 | 40.5 | 39.8 |

## 4.3 ANALYSIS

**Effectiveness of different components.** We validate the effectiveness of different components and design choices for our method in Table 4. As our method is built upon SimGCD, the effectiveness of each component can be observed by comparing its performance with that of SimGCD. We combine SimGCD with the original PatchMix in Zhu et al. (2023) (row 1) as a strong baseline for our task since these are SoTA methods for GCD and UDA respectively. Rows 2-4 indicate our main conceptual methodological contributions. As can be seen, simply combining SimGCD with the original PatchMix developed for UDA leads to a relatively small influence on the results. The original PatchMix focuses mainly on bridging the domain gap of labelled classes through a semi-supervised loss, which limits its capability on the unseen classes from new domains. After subtly adapting PatchMix into contrastive learning for GCD (row 2), the unlabelled data containing both domain shifts and semantic shifts can be properly utilized for training, leading to an obvious performance boost on $\Omega^b$. Furthermore, when we disentangle semantic features from domain features (row 3), the model significantly improves performance on both $\Omega^a$ and $\Omega^b$, demonstrating dissociation of spurious correlations. Appendix E also shows the efficacy of MI regularization in two distinct scenarios. Curriculum sampling further enhances performance on $\Omega^b$ (row 4).

**Incorporating various techniques for GCD with domain shifts.** We study the effectiveness of incorporating the SoTA UDA techniques (MCC and NWD) into the baseline methods. Differently to our task, *domain adaptation does not consider discovering categories in the unlabelled images from the unseen domain*. Results are shown in Table 5. By comparing the results with those of 'Real + Painting' in Table 2, we can see that the results for each method are marginally improved by introducing these techniques. This reveals that simply adopting the UDA techniques to the GCD methods is not sufficient to handle the challenging problem of GCD with domain shifts. Moreover, HiLo again notably outperforms all other methods after introducing these UDA techniques, despite the gain by these techniques being relatively marginal, further demonstrating the effectiveness and significance of our HiLo design. We also extend our analysis by incorporating various UDA techniques (*e.g.*, EFDM Zhang et al. (2022), SFA Li et al. (2021), UniOT Chang et al. (2022)), data augmentation methods (*e.g.*, Mixstyle, Mixup, Cutmix) and curriculum learning (*e.g.*, CL Bengio et al. (2009), SPL Kumar et al. (2010)) into baseline models in Table 6. Our findings reveal that while some UDA techniques and data augmentations offer improvements, they fall short of addressing the full complexity of GCD with domain shifts. Specifically, EFDM improves SimGCD's performance only on $\Omega^a$, likely due to its reliance on explicit source-target domain alignment, which is not

available in our task formulation. The performance of different augmentation methods has a clear drop when compared with our proposed PatchMix for CL (Table 4 row 2). Notably, HiLo consistently outperforms all tested UDA baselines and data augmentation techniques. These results underscore the necessity of our tailored approach for the challenging task of GCD with domain shifts, demonstrating that simply adopting existing UDA or data augmentation methods is insufficient to address this complex problem.

Table 5: Evaluation on the DomainNet dataset by introducing SoTA UDA techniques.

| Methods | | | Real | | | Painting | | |
|---|---|---|---|---|---|---|---|---|
| | | | All | Old | New | All | Old | New |
| RankStats+ | | | 37.3 | 62.1 | 23.4 | 31.0 | **51.2** | 9.2 |
| UNO+ | | | 46.9 | 72.4 | 32.8 | 32.1 | 47.6 | 17.7 |
| ORCA | +MCC+NWD | | 33.4 | 50.1 | 26.7 | 30.0 | 41.1 | 9.1 |
| GCD | | | 50.6 | 54.0 | 48.4 | 34.0 | 43.1 | 22.7 |
| SimGCD | | | 63.1 | 77.1 | 56.9 | 35.7 | 39.0 | 32.4 |
| HiLo (Ours) | | | **65.0** | **77.8** | **58.0** | **42.5** | 43.1 | **42.0** |

Table 6: Influence of different UDA techniques (*e.g.*, Mixstyle, EFDM), data augmentations (*e.g.*, Mixup, Cutmix), and curriculum learning (*e.g.*, CL, SPL). We select the 'Real' and 'Painting' domains from DomainNet to train the DINO model with the SoTA GCD method SimGCD as the baseline and compare with one baseline method UniOT from universal domain adaptation.

| | Methods | Real | | | Painting | | |
|---|---|---|---|---|---|---|---|
| | | All | Old | New | All | Old | New |
| Reference | SimGCD | 61.3 | 77.8 | 52.9 | 34.5 | 35.6 | 33.5 |
| | SimGCD + Mixstyle | 62.3 | 76.8 | 54.0 | 35.0 | 36.1 | 34.0 |
| | SimGCD + EFDM | 62.6 | 76.0 | 54.7 | 34.1 | 34.8 | 33.8 |
| | SimGCD + Mixup | 62.7 | 76.5 | 54.3 | 34.9 | 37.2 | 32.5 |
| | SimGCD + Cutmix | 62.5 | 76.3 | 54.1 | 33.2 | 36.0 | 31.6 |
| Reference | HiLo | 64.4 | 77.6 | 57.5 | 42.1 | 42.9 | 41.3 |
| | HiLo + Mixup - PatchMix | 63.7 | 77.1 | 56.8 | 39.8 | 39.9 | 38.7 |
| | HiLo + Cutmix - PatchMix | 62.9 | 76.8 | 56.0 | 37.4 | 38.0 | 36.7 |
| | HiLo + CL - curriculum sampling | 62.0 | 75.9 | 53.2 | 34.7 | 35.8 | 33.8 |
| | HiLo + SPL - curriculum sampling | 62.8 | 76.5 | 54.5 | 35.0 | 36.1 | 34.0 |
| Reference | HiLo | 64.4 | 77.6 | 57.5 | 42.1 | 42.9 | 41.3 |
| | SFA | 60.1 | 73.4 | 52.8 | 34.9 | 38.0 | 32.6 |
| | UniOT | 63.3 | 77.4 | 57.4 | 35.3 | 38.7 | 32.0 |

**Importance of domain-semantic feature disentanglement.** To validate the necessity of extracting domain and semantic features from different layers, we experiment on two variants of the model, by attaching both heads in $\tilde{\mathcal{H}}$ either to the deepest layer or to the shallowest layer. As shown in rows 5-6 in Table 4, both variants are significantly inferior to our approach using features from different layers. In addition, we further carry out controlled experiments by fixing the layer for one of the two heads while varying the other. In Figure 3 (a), we fix the semantic head to the last layer and vary the 'Shallow' layer for the domain head, from layer 1 to layer 4. As can be seen, attaching the domain head to the earlier layers gives better performance, which also validates that lower-level features are more domain-oriented. Similarly, in Figure 3 (b), we fix the domain head to the first layer and vary the 'Deep' layer for the semantic head, from the last layer to the fourth last layer. We can see that the last layer is the best choice for the semantic head. These results corroborate the importance of domain-semantic feature disentanglement and our design choice of using lower-level features for domain-specific information and higher-level features for semantic semantic-specific information.

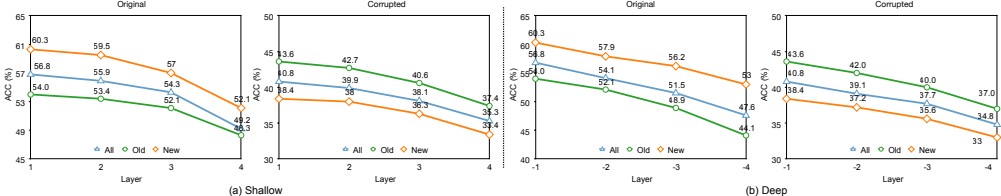

Figure 3: To investigate the effect of features extracted from different layers, we fix the layer for one of the two heads while varying the other on the CUB-C dataset. Features from the first and last layers yield the best performance.

## 5 CONCLUSION

In this paper, we study the new and challenging problem of generalized category discovery under domain shifts. To tackle this challenge, we propose the HiLo learning framework, which contains three major innovations, including domain-semantic disentangled feature learning, PatchMix contrastive learning, and a curriculum learning approach. We thoroughly evaluate HiLo on the DomainNet dataset and our constructed SSB-C benchmark, and show that HiLo outperforms SoTA GCD methods for this challenging problem.

**Acknowledgements** This work is supported by National Natural Science Foundation of China (Grant No. 62306251), Hong Kong Research Grant Council - Early Career Scheme (Grant No. 27208022), and HKU Seed Fund for Basic Research.

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

APPENDIX

# A    SSB-C BENCHMARKS

As demonstrated in Section 4.1 in the main paper, we construct the SSB-C benchmark to evaluate the robustness of algorithms to diverse corruptions applied to validation images of the original SSB benchmark (including CUB Welinder et al. (2010), Stanford Cars Krause et al. (2013a) and FGVC-Aircraft Maji et al. (2013)), adopting the corruptions following Hendrycks & Dietterich (2019). We introduce 9 types of corruption (see Figure 4) in total. Each type of corruption has 5 severity levels. Therefore, SSB-C is $45\times$ larger than the original SSB.

We exclude similar (*i.e.*, defocus blur, glass blur and motion blur) or unrealistic corruptions (*i.e.*, pixel noise and JPEG mosaic). We also exclude corruptions (*i.e.*, bright noise, contrast noise) that may lead to domain leakage (since these corruptions have been adopted during DINO pretraining) to ensure that the model does not see any of the domains in SSB-C during training on the GCD task with domain shifts.

Here is the list of the 9 types of corruption we applied following Hendrycks & Dietterich (2019):

- *Gaussian noise* often appears in low-lighting conditions.
- *Shot noise*, also known as Poisson noise, results from the discrete nature of light and is a form of electronic noise.
- *Impulse noise* occurs due to bit errors and is similar to salt-and-pepper noise but with color variations.
- *Frosted blur* appears on windows or panels with frosted glass texture.
- *Zoom blur* happens when the camera moves rapidly toward an object.
- *Snow* obstructs visibility while frost forms on lenses or windows coated with ice crystals.
- *Fog* shrouds objects and can be rendered using the diamond-square algorithm.
- *Speckle noise* is a granular texture that occurs in coherent imaging systems, such as radar and medical ultrasound. It results from the interference of multiple waves with the same frequency.
- *Spatter* occurs when drops or blobs splash, spot, or soil the images.

Though synthetic, SSB-C incorporates extra challenges and unique values over existing datasets like DomainNet Peng et al. (2019a). Particularly, SSB-C includes (1) fine-grained recognition challenges under domain shifts and (2) more types of domain shifts that are not covered in DomainNet.

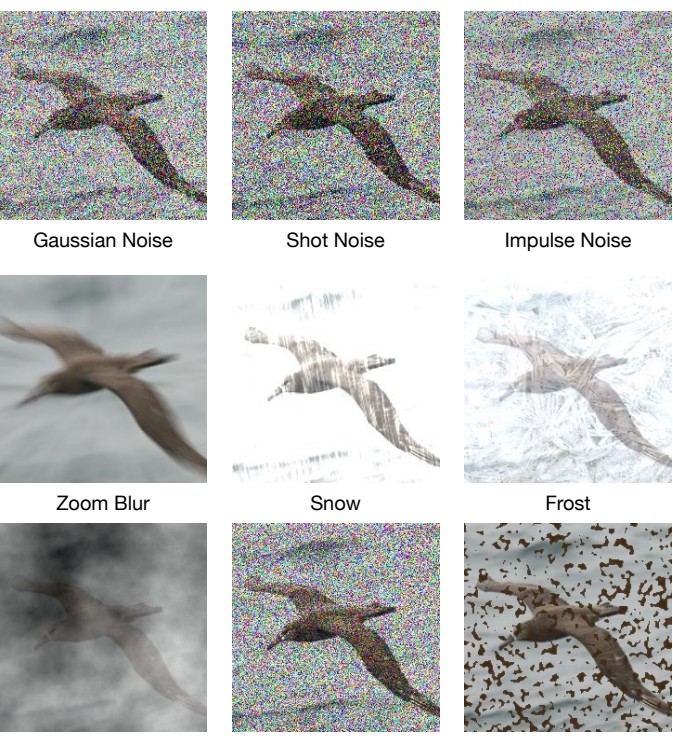

Figure 4: Our SSB-C dataset includes 45 distinct corruptions that are algorithmically generated from 9 types of corruptions, covering *noise*, *blur*, *weather*, and *digital* corruptions. Each type has 5 severity levels.

## B    MULTIPLE UNSEEN DOMAINS FOR DOMAINNET

Due to the large scale of DomainNet (over 587K images), which is significantly larger than SSB-C, it is difficult to utilize all the samples from all remaining domains other than the 'Real' domain in the experiments. Nonetheless, we conduct experiments with multiple domains in DomainNet by subsampling instances while balancing classes and images per class. The total number of unlabelled images from different domains remains the same as the single domain experiment in the paper. Specifically, we randomly select 20% samples from each category in each domain without replacement. Putting all these selected samples from all domains gives a subset which has approximately equivalent number of samples to the total number of samples in 'Painting' domain as in the single domain experiment (see Section 4.2 in the main paper). In this challenging multi-domain experiment, HiLo continues to demonstrate promising results, further validating its effectiveness.

Table 7: Experiments on multiple domains in DomainNet. We subsample instances for the ease of computation, while ensuring class and image balance. The total number of unlabelled images across different domains is kept consistent with the single domain experiment mentioned in the main paper.

| Methods | Real | | | Painting | | | Skecth | | | Quickdraw | | | Clipart | | | Infograph | | |
|---|---|---|---|---|---|---|---|---|---|---|---|---|---|---|---|---|---|---|
| | All | Old | New | All | Old | New | All | Old | New | All | Old | New | All | Old | New | All | Old | New |
| RankStats+ | 34.0 | 62.3 | 19.9 | 30.3 | **50.1** | 11.1 | 17.9 | **31.5** | 7.2 | 2.4 | 2.0 | **2.5** | _25.1_ | **46.4** | 6.3 | 12.0 | **22.1** | 5.5 |
| UNO+ | 43.1 | 72.0 | 28.6 | 30.3 | 43.7 | 17.4 | 12.0 | 16.3 | 8.9 | 2.1 | 2.3 | 1.8 | _22.8_ | _37.4_ | 9.5 | 12.4 | _20.3_ | 6.5 |
| ORCA | 32.1 | 49.9 | 23.5 | 23.0 | 38.8 | 17.0 | 11.6 | 14.7 | 7.6 | _2.8_ | _3.6_ | 2.1 | 20.1 | 33.4 | 10.3 | 8.4 | 17.8 | 6.8 |
| GCD | 47.8 | 53.5 | 45.1 | 32.9 | 40.3 | 26.9 | 17.0 | 22.7 | 11.3 | 1.9 | 2.4 | 1.8 | 24.3 | 31.2 | 15.1 | 10.5 | 12.0 | 9.9 |
| SimGCD | _62.2_ | _77.3_ | _54.3_ | _36.6_ | 42.9 | _30.3_ | _18.2_ | 22.6 | _15.0_ | 2.2 | 2.0 | 2.4 | 25.0 | 34.7 | _16.4_ | _11.8_ | 13.8 | _10.5_ |
| HiLo (Ours) | **65.8** | **77.8** | **58.9** | **43.4** | _49.0_ | **42.9** | **20.0** | _23.6_ | **17.4** | **3.1** | **4.0** | **2.5** | **27.6** | 34.7 | **21.4** | **13.9** | 16.5 | **12.1** |

## C PATCHMIX CONTRASTIVE LEARNING

Specifically, PatchMix consists of a patch embedding layer that transforms input images from labelled and unlabelled data into patches. As outlined in the main paper, PatchMix augments the data by mixing up these patches in the embedding space (as shown in Figure 5 (a)). We randomly sample $\beta$ from Beta distribution to control the proportion of patches from images. Subsequently, we compute the loss for representation learning (Figure 5 (b)) and classification learning (Figure 5 (c)) based on the augmented embeddings and predictions, respectively. The confidence factor $\alpha$ is determined by the overall proportion of known semantics in the mixed samples (*i.e.*, $\beta$ for all the patches) and the attention scores for all the patches of the input image. $\alpha$ is then assigned based on the similarity score or the actual label to guide the training (see Equation (6) in the paper).

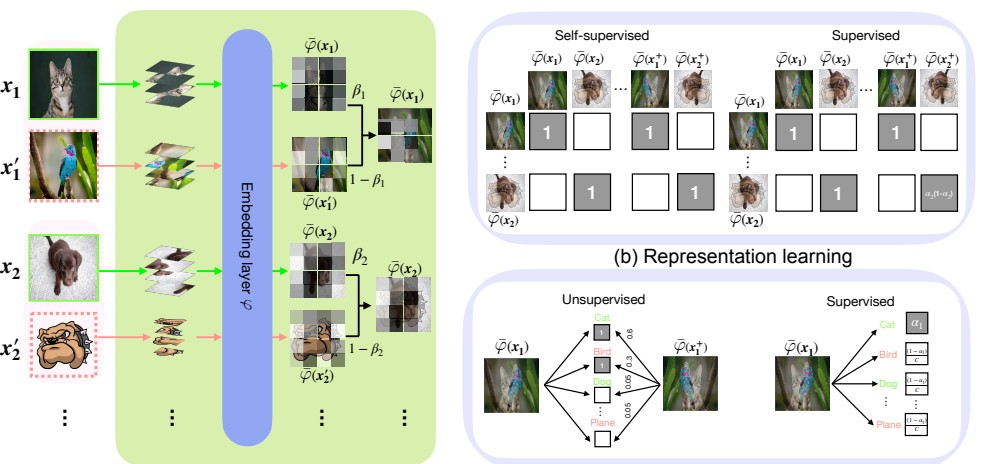

Figure 5: Illustration of PatchMix and loss functions. (a) PatchMix augments the data by mixing up image patches in the embedding space with $\beta$ sampled from Beta distribution. (b) The similarity matrix for representation learning and (c) mixed embedding patches for classification learning are adjusted according to the actual semantic components within the mixed patches, determined by $\alpha$.

# D  THEORETICAL ANALYSIS

Recall that $\mathcal{D}$ is an open-world dataset consisting of a labelled set $\mathcal{D}^l = \{(\boldsymbol{x}_i, y_i)\}_{i=1}^{N_l} \subset \mathcal{X}^l \times \mathcal{Y}^l$ and an unlabelled set $\mathcal{D}^u = \{\boldsymbol{x}_i\}_{i=1}^{N_u} \subset \mathcal{X}^u$. We also define the mapping function $g$ parametrized by a deep neural network as one of the hypotheses from $\mathbb{G}$.

For terminology convenience, here, we term $\mathcal{D}^l$ as the *source domain* data, distributed according to the density $p_s(X, Y)$, while $\mathcal{D}^u$ as the *target domain* data, with a density $p_t(X)$. Note that $\mathcal{D}^u$ contains the unknown mixture of $p_s(X)$ and $p_t(X)$. The objective of our task is to leverage measurable subsets $\Omega^a$ and $\Omega^b$ under $\mathcal{D}_1$ and $\mathcal{D}_2$ to find a hypothesis $g \in \mathbb{G}$ that minimizes the *target error* $e_{\mathcal{D}^u}$, as defined by a zero-one loss function $\ell : \mathcal{Y} \times \mathcal{Y} \to \mathbb{R}$,

$$e_{\mathcal{D}^u}(g) := \mathbb{E}_{x,y \sim \mathcal{D}^u}[\ell(g(x), y)]. \tag{11}$$

More generally, if $y$ is determined by a labelling function $g'$ given the input $x$, we have

$$e_{\mathcal{D}^u}(g, g') := \mathbb{E}_{x \sim \mathcal{D}^u}[\ell(g(x), g'(x))]. \tag{12}$$

Similarly, the *source error* $e_{\mathcal{D}^l}(g)$ and $e_{\mathcal{D}^l}(g, g')$ can be defined by $e_{\mathcal{D}^l}(g) := \mathbb{E}_{x,y \sim \mathcal{D}^l}[\ell(g(x), y)]$ and $e_{\mathcal{D}^l}(g, g') := \mathbb{E}_{x \sim \mathcal{D}^l}[\ell(g(x), g'(x))]$.

When the source domain does not adequately cover the target domain, the target risk of a learned hypothesis cannot be consistently estimated without additional assumptions. Nonetheless, an upper bound on the target risk can be estimated and then minimized. Ben-David et al. (2006) introduce the $\mathcal{A}$-distance (also known as $\mathcal{H}$-divergence) to assess the worst-case loss when extrapolating between domains for hypothesis classes. The $\mathcal{A}$-distance between any two distributions $\mathcal{D}_1$ and $\mathcal{D}_2$ is defined as

$$d_{\mathbb{G}}(\mathcal{D}_1, \mathcal{D}_2) = 2 \sup_{g \in \mathbb{G}} \left| \Pr_{\mathcal{D}_1}[g(x) = 1] - \Pr_{\mathcal{D}_2}[g(x) = 1] \right|.$$

## D.1  PROOF OF BOUNDS FOR THE TARGET ERROR

**Lemma 1.** *Consider a symmetric hypothesis class $\mathbb{G}$ defined on the space $\mathcal{X}$, with a VC dimension $d$. Let $\Omega^a$ and $\Omega^b$ be collections of samples under domains $\mathcal{D}_1$ and $\mathcal{D}_2$. $\hat{d}_{\mathbb{G}}(\Omega^a, \Omega^b)$ is the empirical $\mathcal{A}$-distance between these sample sets. For any $\delta \in (0, 1)$, with probability at least $1 - \delta$,*

$$d_{\mathbb{G}}(\mathcal{D}_1, \mathcal{D}_2) \le 2 \left( 1 - \min_{g \in \mathbb{G}} \left[ \frac{1}{|\Omega^a|} \sum_{x:g(x)=0} \mathbb{1}(x \in \Omega^a) + \frac{1}{|\Omega^b|} \sum_{x:g(x)=1} \mathbb{1}(x \in \Omega^b) \right] \right)$$
$$+ 4 \max \left( \sqrt{\frac{d \log(2|\Omega^a|) - \log \frac{2}{\delta}}{|\Omega^a|}}, \sqrt{\frac{d \log(2|\Omega^b|) - \log \frac{2}{\delta}}{|\Omega^b|}} \right),$$

*Proof.* Recall the definition of the $\mathcal{A}$-distance for hypothesis class $\mathbb{G}$:

$$\hat{d}_{\mathbb{G}}(\Omega^a, \Omega^b) = 2 \sup_{g \in \mathbb{G}} \left| \Pr_{\Omega^a}[I(g)] - \Pr_{\Omega^b}[I(g)] \right|, \tag{13}$$

where

$$\Pr_{\Omega^a}[I(g)] = \frac{1}{|\Omega^a|} \sum_{x \in \Omega^a} \mathbb{1}(g(x) = 1),$$

$$\Pr_{\Omega^b}[I(g)] = \frac{1}{|\Omega^b|} \sum_{x \in \Omega^b} \mathbb{1}(g(x) = 1).$$

For any hypothesis $g$ and corresponding set $I(g)$, we have

$$\Pr_{\Omega^a}[I(g)] - \Pr_{\Omega^b}[I(g)] = \frac{1}{|\Omega^a|} \sum_{x \in \Omega^a} \mathbb{1}(g(x) = 1) - \frac{1}{|\Omega^b|} \sum_{x \in \Omega^b} \mathbb{1}(g(x) = 1).$$

The empirical $\mathcal{A}$-distance is then

$$\hat{d}_{\mathbb{G}}(\Omega^a, \Omega^b) = 2 \sup_{g \in \mathbb{G}} \left| \frac{1}{|\Omega^a|} \sum_{x \in \Omega^a} \mathbb{1}(g(x) = 1) - \frac{1}{|\Omega^b|} \sum_{x \in \Omega^b} \mathbb{1}(g(x) = 1) \right|$$

$$= 2 \sup_{g \in \mathbb{G}} \left| \frac{1}{|\Omega^a|} \sum_{x:g(x)=1} \mathbb{1}(x \in \Omega^a) - \frac{1}{|\Omega^b|} \sum_{x:g(x)=1} \mathbb{1}(x \in \Omega^b) \right|.$$

To simplify this, we consider the complement set where $g(x) = 0$:

$$\Pr_{\Omega^a}[I(g)] - \Pr_{\Omega^b}[I(g)] = \frac{1}{|\Omega^a|} \sum_{x:g(x)=1} \mathbb{1}(x \in \Omega^a) - \frac{1}{|\Omega^b|} \sum_{x:g(x)=1} \mathbb{1}(x \in \Omega^b)$$

$$= 1 - \frac{1}{|\Omega^a|} \sum_{x:g(x)=0} \mathbb{1}(x \in \Omega^a) - \frac{1}{|\Omega^b|} \sum_{x:g(x)=1} \mathbb{1}(x \in \Omega^b).$$

Thus, the empirical $\mathcal{A}$-distance can be expressed as:

$$\hat{d}_{\mathbb{G}}(\Omega^a, \Omega^b) = 2 \sup_{g \in \mathbb{G}} \left| \frac{1}{|\Omega^a|} \sum_{x:g(x)=1} \mathbb{1}(x \in \Omega^a) - \frac{1}{|\Omega^b|} \sum_{x:g(x)=1} \mathbb{1}(x \in \Omega^b) \right|$$

$$= 2 \sup_{g \in \mathbb{G}} \left( 1 - \left( \frac{1}{|\Omega^a|} \sum_{x:g(x)=0} \mathbb{1}(x \in \Omega^a) + \frac{1}{|\Omega^b|} \sum_{x:g(x)=1} \mathbb{1}(x \in \Omega^b) \right) \right).$$

To find the minimum value, we need to consider the complement of the set $I(g)$, which leads to minimizing the expression inside the supremum. This gives us:

$$\hat{d}_{\mathbb{G}}(\Omega^a, \Omega^b) = 2 \left( 1 - \min_{g \in \mathbb{G}} \left( \frac{1}{|\Omega^a|} \sum_{x:g(x)=0} \mathbb{1}(x \in \Omega^a) + \frac{1}{|\Omega^b|} \sum_{x:g(x)=1} \mathbb{1}(x \in \Omega^b) \right) \right).$$

From Theorem 3.4 of Kifer et al. (2004), we can know that:

$$P^{|\Omega^a|+|\Omega^b|} \left[ |\hat{d}_{\mathbb{G}}(\Omega^a, \Omega^b) - d_{\mathbb{G}}(\mathcal{D}_1, \mathcal{D}_2)| > \epsilon \right] \le (2|\Omega^a|)^d e^{-|\Omega^a|\epsilon^2/16} + (2|\Omega^b|)^d e^{-|\Omega^b|\epsilon^2/16}$$

$$= \delta.$$

We use a union bound to handle the two terms separately:

$$(2m)^d e^{-|\Omega^a|\epsilon^2/16} \le \frac{\delta}{2} \quad \text{and} \quad (2|\Omega^b|)^d e^{-|\Omega^b|\epsilon^2/16} \le \frac{\delta}{2}$$

For the first inequality:

$$(2|\Omega^a|)^d e^{-|\Omega^a|\epsilon^2/16} \le \frac{\delta}{2}$$

Taking the natural logarithm on both sides:

$$\log((2|\Omega^a|)^d) - \frac{|\Omega^a|\epsilon^2}{16} \le \log \frac{\delta}{2}$$

$$d \log(2|\Omega^a|) - \frac{|\Omega^a|\epsilon^2}{16} \le \log \frac{\delta}{2}$$

$$\epsilon^2 \ge \frac{16}{|\Omega^a|} \left( d \log(2|\Omega^a|) - \log \frac{2}{\delta} \right)$$

$$\epsilon \ge 4 \sqrt{\frac{d \log(2|\Omega^a|) - \log \frac{2}{\delta}}{|\Omega^a|}}$$

Similarly, for the second inequality:

$$\epsilon \geq 4\sqrt{\frac{d\log(2|\Omega^b|) - \log\frac{2}{\delta}}{|\Omega^b|}}$$

To ensure that both inequalities hold, we take the maximum of the two derived $\epsilon$ values:

$$\epsilon \geq \max\left(4\sqrt{\frac{d\log(2|\Omega^a|) - \log\frac{2}{\delta}}{|\Omega^a|}}, 4\sqrt{\frac{d\log(2|\Omega^b|) - \log\frac{2}{\delta}}{|\Omega^b|}}\right)$$

Thus, we have

$$d_{\mathbb{G}}(\mathcal{D}_1, \mathcal{D}_2) \leq \hat{d}_{\mathbb{G}}(\Omega^a, \Omega^b) + 4\max\left(\sqrt{\frac{d\log(2|\Omega^a|) - \log\frac{2}{\delta}}{|\Omega^a|}}, \sqrt{\frac{d\log(2|\Omega^b|) - \log\frac{2}{\delta}}{|\Omega^b|}}\right)$$

$$= 2\left(1 - \min_{g \in \mathbb{G}}\left[\frac{1}{|\Omega^a|}\sum_{x:g(x)=0}\mathbb{1}(x \in \Omega^a) + \frac{1}{|\Omega^b|}\sum_{x:g(x)=1}\mathbb{1}(x \in \Omega^b)\right]\right)$$

$$+ 4\max\left(\sqrt{\frac{d\log(2|\Omega^a|) - \log\frac{2}{\delta}}{|\Omega^a|}}, \sqrt{\frac{d\log(2|\Omega^b|) - \log\frac{2}{\delta}}{|\Omega^b|}}\right),$$

$\square$

**Theorem 1.** *Consider a symmetric hypothesis class $\mathbb{G}$ defined on the space $\mathcal{X}$, with a VC dimension $d$. Let $\Omega^a$ and $\Omega^b$ be collections of samples under domains $\mathcal{D}_1$ and $\mathcal{D}_2$. For any $\delta \in (0,1)$, with probability at least $1 - \delta$,*

$$e_{\mathcal{D}^u}(g) \leq e_{\mathcal{D}^l}(g) + \left(1 - \min_{g \in \mathbb{G}}\left[\frac{1}{|\Omega^a|}\sum_{x:g(x)=0}\mathbb{1}(x \in \Omega^a) + \frac{1}{|\Omega^b|}\sum_{x:g(x)=1}\mathbb{1}(x \in \Omega^b)\right]\right)$$

$$+ 2\max\left(\sqrt{\frac{d\log(2|\Omega^a|) - \log\frac{2}{\delta}}{|\Omega^a|}}, \sqrt{\frac{d\log(2|\Omega^b|) - \log\frac{2}{\delta}}{|\Omega^b|}}\right),$$

*Proof.* Let $g \in \mathbb{G}$ be a hypothesis such that $g(x) = 1$ if and only if $g_1(x) \neq g_2(x)$ for some $g_1, g_2 \in \mathbb{G}$, indicating a disagreement between $g_1(x)$ and $g_2(x)$. Based on the definition of $\mathcal{A}$-distance, we have

$$d_{\mathbb{G}}(\mathcal{D}_1, \mathcal{D}_2) = 2\sup_{g \in \mathbb{G}}\left|\Pr_{\mathcal{D}_1}[g(x) = 1] - \Pr_{\mathcal{D}_2}[g(x) = 1]\right|$$

$$= 2\sup_{g_1, g_2 \in \mathbb{G}}|e_{\mathcal{D}_1}(g_1, g_2) - e_{\mathcal{D}_2}(g_1, g_2)|$$

$$= 2|e_{\mathcal{D}_1}(g_1, g_2) - e_{\mathcal{D}_2}(g_1, g_2)|.$$

Consider an ideal joint hypothesis $g^*$, which is the hypothesis which minimizes the combined error (ideally zero). By using the triangle inequality Ben-David et al. (2006), we have:

$$d_{\mathbb{G}}(\mathcal{D}_1, \mathcal{D}_2) \geq 2|(e_{\mathcal{D}_1}(g_1) - e_{\mathcal{D}_1}(g^*)) - (e_{\mathcal{D}_2}(g_1) - e_{\mathcal{D}_2}(g^*))|$$

$$\geq 2|e_{\mathcal{D}_1}(g_1) - e_{\mathcal{D}_2}(g_1)|.$$

As $\mathcal{D}^u$ contains samples from both $\mathcal{D}_1$ and $\mathcal{D}_2$, we immediately know that:

$$d_{\mathbb{G}}(\mathcal{D}^l, \mathcal{D}^u) \leq d_{\mathbb{G}}(\mathcal{D}_1, \mathcal{D}_2)$$

By Lemma 1, we have that

$$2(e_{\mathcal{D}^u}(g) - e_{\mathcal{D}^l}(g)) \leq 2\left(1 - \min_{g \in \mathbb{G}}\left[\frac{1}{|\Omega^a|}\sum_{x:g(x)=0}\mathbb{1}(x \in \Omega^a) + \frac{1}{|\Omega^b|}\sum_{x:g(x)=1}\mathbb{1}(x \in \Omega^b)\right]\right)$$

$$+ 4\max\left(\sqrt{\frac{d\log(2|\Omega^a|) - \log\frac{2}{\delta}}{|\Omega^a|}}, \sqrt{\frac{d\log(2|\Omega^b|) - \log\frac{2}{\delta}}{|\Omega^b|}}\right)$$

$$e_{\mathcal{D}^u}(g) \leq e_{\mathcal{D}^l}(g) + \left(1 - \min_{g \in \mathbb{G}}\left[\frac{1}{|\Omega^a|}\sum_{x:g(x)=0}\mathbb{1}(x \in \Omega^a) + \frac{1}{|\Omega^b|}\sum_{x:g(x)=1}\mathbb{1}(x \in \Omega^b)\right]\right)$$

$$+ 2\max\left(\sqrt{\frac{d\log(2|\Omega^a|) - \log\frac{2}{\delta}}{|\Omega^a|}}, \sqrt{\frac{d\log(2|\Omega^b|) - \log\frac{2}{\delta}}{|\Omega^b|}}\right).$$

$\square$

Theorem 1 demonstrates that the upper bound of the error on $\mathcal{D}^u$ depends on the error on $\mathcal{D}^l$ and the domain classification performance of $g$. It is evident that all components involved in Equation (8) minimize the misclassification error in the second item. For the first item, curriculum sampling ensures a reduced error of $g$ on $\mathcal{D}^l$ during the early training stage, before HiLo can accurately classify different domains through the domain head (thus leading to a lower error in domain classification, *i.e.*, the second item).

## D.2 A TIGHTER BOUND FOR THE TARGET ERROR

**Lemma 2.** *For the hypothesis class $\mathbb{G}$,*

$$d_{\mathbb{G}}(\mathcal{D}_1, \mathcal{D}_2) \leq 2\|\mathcal{D}_1 - \mathcal{D}_2\|_{TV}.$$

*Proof.* Recall that the total variation (TV) distance between two distributions $\mathcal{D}_1$ and $\mathcal{D}_2$ is defined as:

$$\|\mathcal{D}_1 - \mathcal{D}_2\|_{TV} = \sup_A |\mathcal{D}_1(A) - \mathcal{D}_2(A)|,$$

where the supremum is taken over all measurable sets $A$.

The $\mathcal{A}$-distance can be seen as a specific form of the TV distance where the measurable sets $A$ are the subsets of the input space that can be defined by the hypotheses $g \in \mathbb{G}$. However, the TV distance considers all possible measurable sets $A$. For any measurable set $A$, we can consider the indicator function $\mathbb{1}_A(x)$ which takes value 1 if $x \in A$ and 0 otherwise. The TV distance can be expressed in terms of these indicator functions:

$$\|\mathcal{D}_1 - \mathcal{D}_2\|_{TV} = \sup_A |\mathcal{D}_1(A) - \mathcal{D}_2(A)| = \sup_A \left|\int \mathbb{1}_A(x)\,d\mathcal{D}_1(x) - \int \mathbb{1}_A(x)\,d\mathcal{D}_2(x)\right|.$$

When considering the hypothesis class $\mathbb{G}$, we look at the functions $g(x)$ that take the value 1 or 0, similar to indicator functions for sets:

$$d_{\mathbb{G}}(\mathcal{D}_1, \mathcal{D}_2) = 2\sup_{g \in \mathbb{G}}\left|\Pr_{\mathcal{D}_1}[g(x) = 1] - \Pr_{\mathcal{D}_2}[g(x) = 1]\right|.$$

For a given hypothesis $g \in \mathbb{G}$, let $A_g = \{x \mid g(x) = 1\}$. The difference in probabilities for this hypothesis is:

$$\left|\Pr_{\mathcal{D}_1}[g(x) = 1] - \Pr_{\mathcal{D}_2}[g(x) = 1]\right| = |\mathcal{D}_1(A_g) - \mathcal{D}_2(A_g)|.$$

We left off by noting that for any hypothesis $g \in \mathbb{G}$, the difference in probabilities $|\Pr_{\mathcal{D}_1}[g(x) = 1] - \Pr_{\mathcal{D}_2}[g(x) = 1]|$ is bounded by the TV distance $\|\mathcal{D}_1 - \mathcal{D}_2\|_{TV}$:

$$|\mathcal{D}_1(A_g) - \mathcal{D}_2(A_g)| \leq \|\mathcal{D}_1 - \mathcal{D}_2\|_{TV},$$

where $A_g = \{x \mid g(x) = 1\}$.

The $\mathcal{A}$-distance takes the supremum of this difference over all hypotheses $g \in \mathbb{G}$:

$$d_{\mathbb{G}}(\mathcal{D}_1, \mathcal{D}_2) = 2 \sup_{g \in \mathbb{G}} \left| \Pr_{\mathcal{D}_1}[g(x) = 1] - \Pr_{\mathcal{D}_2}[g(x) = 1] \right|.$$

Because each individual difference $|\Pr_{\mathcal{D}_1}[g(x) = 1] - \Pr_{\mathcal{D}_2}[g(x) = 1]|$ is bounded by $\|\mathcal{D}_1 - \mathcal{D}_2\|_{TV}$, the supremum over all such differences must also be bounded by $\|\mathcal{D}_1 - \mathcal{D}_2\|_{TV}$:

$$\sup_{g \in \mathbb{G}} \left| \Pr_{\mathcal{D}_1}[g(x) = 1] - \Pr_{\mathcal{D}_2}[g(x) = 1] \right| \leq \|\mathcal{D}_1 - \mathcal{D}_2\|_{TV}$$

$$2 \sup_{g \in \mathbb{G}} \left| \Pr_{\mathcal{D}_1}[g(x) = 1] - \Pr_{\mathcal{D}_2}[g(x) = 1] \right| \leq 2\|\mathcal{D}_1 - \mathcal{D}_2\|_{TV}$$

$$d_{\mathbb{G}}(\mathcal{D}_1, \mathcal{D}_2) \leq 2\|\mathcal{D}_1 - \mathcal{D}_2\|_{TV}.$$

$\square$

**Lemma 3.** *Let a random variable $z \in \mathcal{Z}$ be a representation of the input features $X$. $\mathcal{F}_\varphi(X) =: z$ with $\mathcal{F}_\varphi \in \mathbb{F}$ is a feature transformation and $\mathcal{H} \in \mathbb{H}$ operating in the representation space $\mathcal{Z}$ is a prediction function. Hypotheses $g \in \mathbb{G}$ are formed by compositions $g = \mathcal{H} \circ \mathcal{F}_\varphi$ and $\mathbb{G} := \{\mathcal{H} \circ \mathcal{F}_\varphi : H \in \mathbb{H}, \mathcal{F}_\varphi \in \mathbb{F}\}$. For all $\mathcal{F}_\varphi \in \mathbb{F}$ and $\mathcal{H} \in \mathbb{H}$,*

$$d_{\mathbb{H}}(\mathcal{Z}^l, \mathcal{Z}^u) \leq d_{\mathbb{G}}(\mathcal{D}^l, \mathcal{D}^u).$$

*Proof.* Let $g \in \mathbb{G}$ be a hypothesis such that $g(x) = 1$ if and only if $g_1(x) \neq g_2(x)$ for some $g_1, g_2 \in \mathbb{G}$, indicating a disagreement between $g_1(x)$ and $g_2(x)$. Based on the definition of $\mathcal{A}$-distance, we have

$$d_{\mathbb{G}}(\mathcal{D}^l, \mathcal{D}^u) = 2 \sup_{g \in \mathbb{G}} \left| \Pr_{\mathcal{D}^l}[g(x) = 1] - \Pr_{\mathcal{D}^u}[g(x) = 1] \right|,$$

and similarly,

$$d_{\mathbb{H}}(\mathcal{Z}^d, \mathcal{Z}^s) = 2 \sup_{\mathcal{H} \in \mathbb{H}} \left| \Pr_{\mathcal{Z}^d}[\mathcal{H}(z) = 1] - \Pr_{\mathcal{Z}^s}[\mathcal{H}(z) = 1] \right|.$$

For $Z = \mathcal{F}_\varphi(X)$, we know that:

$$\Pr_{z \sim \mathcal{Z}^l}[\mathcal{H}(z) = 1] = \Pr_{x \sim \mathcal{D}^l}[\mathcal{H}(\mathcal{F}_\varphi(x)) = 1],$$

and

$$\Pr_{z \sim \mathcal{Z}^u}[\mathcal{H}(z) = 1] = \Pr_{x \sim \mathcal{D}^u}[\mathcal{H}(\mathcal{F}_\varphi(x)) = 1].$$

For each $\mathcal{H} \in \mathbb{H}$, there is a corresponding $g \in \mathbb{G}$ such that $g = \mathcal{H} \circ \mathcal{F}_\varphi$. However, not every $g \in \mathbb{G}$ has a corresponding $\mathcal{H} \in \mathbb{H}$ because $\mathcal{F}_\varphi$ might not cover the entire space or map back uniquely. This leads to:

$$\sup_{\mathcal{H} \in \mathbb{H}} \left| \Pr_{z \sim \mathcal{Z}^l}[\mathcal{H}(z) = 1] - \Pr_{z \sim \mathcal{Z}^u}[\mathcal{H}(z) = 1] \right| \leq \sup_{g \in \mathbb{G}} \left| \Pr_{x \sim \mathcal{D}^l}[g(x) = 1] - \Pr_{x \sim \mathcal{D}^u}[g(x) = 1] \right|$$

$$2 \sup_{\mathcal{H} \in \mathbb{H}} \left| \Pr_{z \sim \mathcal{Z}^l}[\mathcal{H}(z) = 1] - \Pr_{z \sim \mathcal{Z}^u}[\mathcal{H}(z) = 1] \right| \leq 2 \sup_{g \in \mathbb{G}} \left| \Pr_{x \sim \mathcal{D}^l}[g(x) = 1] - \Pr_{x \sim \mathcal{D}^u}[g(x) = 1] \right|$$

$$d_{\mathbb{H}}(\mathcal{Z}^l, \mathcal{Z}^u) \leq d_{\mathbb{G}}(\mathcal{D}^l, \mathcal{D}^u).$$

$\square$

**Lemma 4.** *Let a random variable $z \in \mathcal{Z}$ be a representation of the input features $X$. $\psi(z) =: z_d, z_s$ with $\psi \in \Psi$ is a separator for the domain-specific feature $z_d \in \mathcal{Z}^d$ and the semantic-specific feature $z_s \in \mathcal{Z}^s$. Let $\mathcal{J} \in \mathbb{J}$ be a prediction function based on $\mathcal{Z}^d, \mathcal{Z}^s$ and $I(z_d; z_s)$ be the mutual information between $z_d$ and $z_s$, for a hypothesis space $\mathbb{J}$,*

$$d_{\mathbb{J}}(\mathcal{Z}^d, \mathcal{Z}^s) \le d_{\mathbb{H}}(\mathcal{Z}^l, \mathcal{Z}^u) \le d_{\mathbb{G}}(\mathcal{D}^l, \mathcal{D}^u).$$

*Proof.* Hypotheses $g \in \mathbb{G}$ can be formed by either compositions $g = \mathcal{H} \circ \mathcal{F}_\varphi$ and $\mathbb{G} := \{\mathcal{H} \circ \mathcal{F}_\varphi : H \in \mathbb{H}, \mathcal{F}_\varphi \in \mathbb{F}\}$, or compositions $g = \mathcal{J} \circ \psi \circ \mathcal{F}_\varphi$ and $\mathbb{G} := \{\mathcal{J} \circ \psi \circ \mathcal{F}_\varphi : \mathcal{J} \in \mathbb{J}, \psi \in \Psi, \mathcal{F}_\varphi \in \mathbb{F}\}$. Then, similar to the proof for Lemma 3, we can easily get the inequality.

$\square$

**Theorem 2.** *Let $\mathbb{G}$ be a symmetric hypothesis class defined on the space $\mathcal{X}$, with a VC dimension $d$. Let $\Omega^a$ and $\Omega^b$ be collections of samples under domains $\mathcal{D}_1$ and $\mathcal{D}_2$, $z_d$ and $z_s$ be drawn from $\mathcal{Z}^d$ and $\mathcal{Z}^s$, and $I(z_d; z_s)$ be the mutual information between $z_d$ and $z_s$. Define the optimal function $g^* \in \mathbb{G}$ as follows:*

$$g^* = \arg\min_{g \in \mathbb{G}} \left[ \frac{1}{|\Omega^a|} \sum_{x: g(x)=0} \mathbb{1}(x \in \Omega^a) + \frac{1}{|\Omega^b|} \sum_{x: g(x)=1} \mathbb{1}(x \in \Omega^b) \right].$$

*Then, $e_{\mathcal{D}^u}(g)$ is more tightly bounded by $e_{\mathcal{D}^l}(g) + \sqrt{I(z_d; z_s)}$.*

*Proof.* Recall that $\mathcal{A}$-distance between two distributions $\mathcal{Z}^d$ and $\mathcal{Z}^s$ is defined as:

$$d_{\mathbb{J}}(\mathcal{Z}^d, \mathcal{Z}^s) = 2 \sup_{g \in \mathbb{G}} \left| \Pr_{\mathcal{Z}^d}[g(z) = 1] - \Pr_{\mathcal{Z}^s}[g(z) = 1] \right|.$$

Recall that the Jensen-Shannon (JS) divergence between two distributions $\mathcal{Z}^d$ and $\mathcal{Z}^s$ is defined as:

$$\mathcal{D}_{JS}(\mathcal{Z}^d \| \mathcal{Z}^s) = \frac{1}{2} \left( \mathcal{D}_{KL}(\mathcal{Z}^d \| M) + \mathcal{D}_{KL}(\mathcal{Z}^s \| M) \right),$$

where $M = \frac{1}{2}(\mathcal{Z}^d + \mathcal{Z}^s)$ is the mixture distribution and $\mathcal{D}_{KL}$ is the Kullback-Leibler (KL) divergence. Using Pinsker's Inequality Csiszár & Körner (2011), we have:

$$\|\mathcal{Z}^d - \mathcal{Z}^s\|_{TV}^2 \le \frac{1}{2} \left( \mathcal{D}_{KL}(\mathcal{Z}^d \| M) + \mathcal{D}_{KL}(\mathcal{Z}^s \| M) \right).$$

This implies:

$$\|\mathcal{Z}^d - \mathcal{Z}^s\|_{TV} \le \sqrt{\frac{1}{2} \left( \mathcal{D}_{KL}(\mathcal{Z}^d \| M) + \mathcal{D}_{KL}(\mathcal{Z}^s \| M) \right)}.$$

Thus, we can rewrite the TV distance bound in terms of the JS divergence:

$$\|\mathcal{Z}^d - \mathcal{Z}^s\|_{TV} \le \sqrt{\mathcal{D}_{JS}(\mathcal{Z}^d \| \mathcal{Z}^s)}.$$

From Lemma 2, we know that the $\mathcal{A}$-distance is bounded by twice the TV distance:

$$d_{\mathbb{J}}(\mathcal{Z}^d, \mathcal{Z}^s) \le 2\|\mathcal{Z}^d - \mathcal{Z}^s\|_{TV}.$$

Using the bound on the TV distance in terms of the JS divergence, we get:

$$d_{\mathbb{J}}(\mathcal{Z}^d, \mathcal{Z}^s) \le 2\sqrt{\mathcal{D}_{JS}(\mathcal{Z}^d \| \mathcal{Z}^s)}.$$

As Equation (4) is Donsker-Varadhan representations of KL divergence Donsker & Varadhan (1983) that approximates the mutual information using a neural network (MLP) $\Phi$, the bound on mutual information in terms of the JS divergence is:

$$
\begin{aligned}
I(\boldsymbol{z}_d; \boldsymbol{z}_s) &= \mathcal{D}_{KL}(\mathcal{Z}^d \| \mathcal{Z}^s) \\
&\geq \frac{1}{2}\mathcal{D}_{KL}(\mathcal{Z}^d \| M) + \frac{1}{2}\mathcal{D}_{KL}(\mathcal{Z}^s \| M) \quad (\text{Define } M = \frac{1}{2}(\mathcal{Z}^d + \mathcal{Z}^s)) \\
&= \mathcal{D}_{JS}(\mathcal{Z}^d \| \mathcal{Z}^s).
\end{aligned}
$$

Substituting this into the $\mathcal{A}$-distance bound, we get:

$$
d_{\mathbb{J}}(\mathcal{Z}^d, \mathcal{Z}^s) \leq 2\sqrt{\mathcal{D}_{JS}(\mathcal{Z}^d \| \mathcal{Z}^s)} \leq 2\sqrt{I(z_d; z_s)}.
$$

Given that $g^*$ is the optimal function Given that $g^*$ is the optimal function from the set of hypotheses $\mathbb{G}$, we consider the bound of $\hat{d}_{\mathbb{G}}(\Omega^a, \Omega^b)$ stated in Lemma 1. Specifically, we have:

$$
\begin{aligned}
\hat{d}_{\mathbb{G}}(\Omega^a, \Omega^b) \leq 2\Bigg( & 1 - \min_{g \in \mathbb{G}} \left[ \frac{1}{|\Omega^a|} \sum_{x:g(x)=0} \mathbb{1}(x \in \Omega^a) + \frac{1}{|\Omega^b|} \sum_{x:g(x)=1} \mathbb{1}(x \in \Omega^b) \right] \Bigg) \\
& + 4\max\left( \sqrt{\frac{d\log(2|\Omega^a|) - \log\frac{2}{\delta}}{|\Omega^a|}}, \sqrt{\frac{d\log(2|\Omega^b|) - \log\frac{2}{\delta}}{|\Omega^b|}} \right),
\end{aligned}
$$

The minimum of this bound cannot be lower than 4. Considering the mutual information $I(z_d; z_s)$, we know that:

$$
0 \leq I(z_d; z_s) \leq 1.
$$

Therefore, it follows that:

$$
0 \leq 2\sqrt{I(z_d; z_s)} \leq 2.
$$

This implies that $2\sqrt{I(z_d; z_s)}$ can serve as a tighter upper bound for $\hat{d}_{\mathbb{G}}(\Omega^a, \Omega^b)$.

Replacing the tighter bound of $\hat{d}_{\mathbb{G}}(\Omega^a, \Omega^b)$ in the proof steps of Theorem 1, we have

$$
\begin{aligned}
2(e_{\mathcal{D}^u}(g) - e_{\mathcal{D}^l}(g)) &\leq 2\sqrt{I(z_d; z_s)} \\
e_{\mathcal{D}^u}(g) &\leq e_{\mathcal{D}^l}(g) + \sqrt{I(z_d; z_s)}.
\end{aligned}
$$

$\square$

# E EFFECT OF MUTUAL INFORMATION MINIMIZATION ON DIFFERENT DATASETS

Our comprehensive analysis of HiLo's performance across diverse datasets, with and without Mutual Information (MI) regularization, is presented in Figure 6. The results demonstrate that MI regularization's efficacy is particularly pronounced in two distinct scenarios: datasets with markedly distinct low-level styles (*e.g.*, Quickdraw, Infograph) and those with closely related semantic categories (*e.g.*, SSB-C benchmark). In the case of Quickdraw, the dramatic style variations are readily captured by low-level features, allowing MI regularization to effectively disentangle semantic features from low-level information. Note that Quickdraw samples are still too abstract to learn from constrastive learning, leading to poor performance of the model. For the SSB-C benchmark, where image structure remains largely unchanged across different noise types, MI regularization proves crucial in distinguishing subtle semantic differences from low-level information variations.

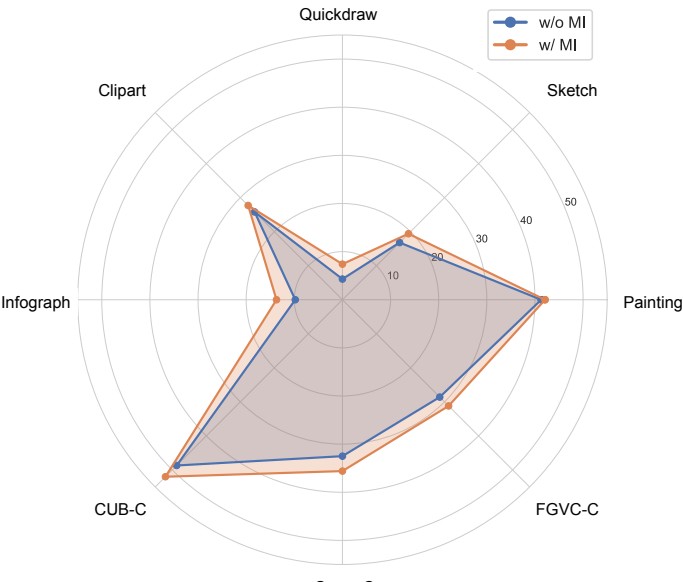

Figure 6: Comparison between HiLo with and without MI across different datasets. DomainNet is a generic dataset with disparate styles (*e.g.*, Painting, Quickdraw, Sketch, Clipart and Infograph). The labelled data are all from 'Real' domain. SSB-C (*e.g.*, CUB-C, Scars-C and FGVC-C) is created by adding several common noises in the real world to the fine-grained datasets. The labelled data are all from the original SSB. When low-level style is quite different (*e.g.*, Infograph, Quickdraw) or semantics are close (*e.g.*, SSB-C benchmark), the improvement of MI regularization is pronounced.

## F  NOVEL CATEGORY DISCOVERY IN THE PRESENCE OF DOMAIN SHIFTS

In this paper, we consider a challenging problem of generalized category discovery with domain shifts, which, to our knowledge, has not been studied in the literature. However, the study of novel category discovery under domain shifts has been considered in Yu et al. (2022) from a domain adaptation perspective, which introduces a self-labeling framework, called NCDD, that can categorize unlabelled images from both source and target domains, by *maximizing* mutual information between labels and input images. The unlabelled images from the target domain may contain images from new categories that are not present in the source domain. Differently, in our study, we consider that unseen classes are also present in the unlabelled data from the source domain (*i.e.*, the domain $\Omega^a$), and new domains may appear at test time. Meanwhile, our HiLo framework also differs significantly from NCDD. Particularly, HiLo learns to disentangle domain-semantic features by *minimizing* the mutual information between domain and semantic heads. It also incorporates a novel PatchMix contrastive learning method and a curriculum learning approach to facilitate the robustness of representation to domain shifts. To compare HiLo with NCDD, we reimplement the NCDD method[1] and experiment on the experimental configuration following Yu et al. (2022) on the CUB-C. We present the results in Table 8. As can be seen, HiLo significantly outperforms NCDD and all other baselines, highlighting its effectiveness on domain-semantic disentanglement.

Table 8: Evaluation on SSB-C datasets. We report results of baselines in the seen domain (*i.e.*, Original) and the overall performance of different corruptions (*i.e.*, Corrupted).

| | CUB-C | | | | | |
| | Original | | | Corrupted | | |
| Methods | All | Old | New | All | Old | New |
| --- | --- | --- | --- | --- | --- | --- |
| RankStats+ | 19.3 | 22.0 | 15.4 | 13.6 | 23.9 | 4.5 |
| UNO+ | 25.9 | 40.1 | 21.3 | 21.5 | 33.4 | 8.6 |
| ORCA | 18.2 | 22.8 | 14.5 | 21.5 | 23.1 | 18.9 |
| NCDD | 37.0 | 50.7 | 28.7 | 30.2 | 53.0 | 11.7 |
| GCD | 26.6 | 27.5 | 25.7 | 25.1 | 28.7 | 22.0 |
| SimGCD | 31.9 | 33.9 | 29.0 | 28.8 | 31.6 | 25.0 |
| HiLo (Ours) | **56.8** | **54.0** | **60.3** | **52.0** | **53.6** | **50.5** |

---

[1]Our NCDD reimplementation's performance aligns with other efforts An et al. (2023)

## G   INVESTIGATION OF CLIP FOR GCD WITH DOMAIN SHIFTS

CLIP Radford et al. (2021) has demonstrated strong performance in various computer vision tasks. We thus investigate its potential for the challenging problem of GCD with domain shifts. We employ the pretrained vision transformer from CLIP as the backbone for HiLo. As illustrated in Table 9, employing CLIP significantly improves the performance of HiLo on DomainNet, compared with the DINO-based HiLo and SimGCD.

Table 9: Effectiveness of employing CLIP as the backbone for HiLo. We select the 'Real' and 'Painting' domains to train the DINO model with the techniques introduced above as the baseline.

|  | | Real | | | Painting | | |
| --- | --- | --- | --- | --- | --- | --- | --- |
|  | Methods | All | Old | New | All | Old | New |
| Baseline | SimGCD Wen et al. (2023) | 61.3 | 77.8 | 52.9 | 34.5 | 35.6 | 33.5 |
|  | + Pretrained CLIP | 69.8 | 77.2 | 58.9 | 37.1 | 38.0 | 35.1 |
| Baseline | HiLo | 64.4 | 77.6 | 57.5 | 42.1 | 42.9 | 41.3 |
|  | + Pretrained CLIP | 74.5 | 78.1 | 64.2 | 47.1 | 49.5 | 45.4 |

Table 9, verifies that a strong visual encoder can bring performance boost on both seen and novel domains. In Table 10, we further compare with another two CLIP baselines to better understand the potential of the visual language model, *i.e.*, zero-shot CLIP with oracle class names (which are not expected to be unavailable in GCD) and with zero-shot CLIP a very large vocabulary (*i.e.*, WordNet Miller (1995)), where we conduct zero-shot inference using the class names of both known and unknown classes, by comparing the visual feature of each image and the text features of class descriptions. The results in Table 10 demonstrate that CLIP models do not enhance robustness compared to our HiLo model, a visual-only model, on CUB-C, despite that an extra vocabulary is provided for the CLIP model (which arguably reduces the difficulty of the GCD task, in which we do not assume any extra textual or visual knowledge on the unlabelled data). This finding aligns with Taori et al. (2020), which indicates that robustness under natural distribution shifts does not necessarily translate to robustness under synthetic distribution shifts, thereby suggesting the limited impact of CLIP models on covariate shifts. HiLo outperforms CLIP[†] and CLIP[‡], despite that they 'cheat' by using an extra vocabulary, which further underscores the effectiveness and robustness of HiLo and the challenge of GCD with domain shifts.

Table 10: Zero-shot performance of CLIP on CUB. CLIP[†] is the CLIP with oracle class names, while CLIP[‡] is the CLIP with a large vocabulary (*i.e.*, WordNet Miller (1995)).

|  | CUB-C | | | | | |
| --- | --- | --- | --- | --- | --- | --- |
|  | Original | | | Corrupted | | |
| Methods | All | Old | New | All | Old | New |
| RankStats+ | 19.3 | 22.0 | 15.4 | 13.6 | 23.9 | 4.5 |
| UNO+ | 25.9 | 40.1 | 21.3 | 21.5 | 33.4 | 8.6 |
| ORCA | 18.2 | 22.8 | 14.5 | 21.5 | 23.1 | 18.9 |
| GCD | 26.6 | 27.5 | 25.7 | 25.1 | 28.7 | 22.0 |
| SimGCD | 31.9 | 33.9 | 29.0 | 28.8 | 31.6 | 25.0 |
| HiLo (Ours) | **56.8** | **54.0** | **60.3** | **52.0** | **53.6** | **50.5** |
| CLIP[†] | *55.5* | *51.6* | *57.4* | *50.3* | *51.8* | *48.9* |
| CLIP[‡] | *55.1* | *51.0* | *57.1* | *49.6* | *51.4* | *47.8* |

# H   DETAILED EVALUATION OF SSB-C DATASETS

In addition to the overall SSB-C results presented in the main paper, we provide a detailed analysis of CUB-C, Scars-C, and FGVC-C against various corruptions in Table 12 and Table 13. Our proposed method consistently outperforms the baselines. Notably, while Gaussian, Speckle, Impulse, and Shot noise corruptions appear qualitatively similar, their performance impacts differ significantly. Specifically, Speckle noise has a less detrimental effect on performance compared to other noise types. As illustrated in Figure 4, Speckle noise preserves more semantic information, whereas other noises pervade the images. This retention of semantic information is crucial for accurate object recognition in fine-grained settings, explaining the consistently better performance on Speckle noise compared to other corruption types.

Table 11: A detailed evaluation of the CUB-C dataset. We assess the performance of each individual corruption.

| Methods | Gaussian Noise | | | Shot Noise | | | Impulse Noise | | | Zoom Blur | | | Snow | | | Frost | | | Fog | | | Speckle | | | Spatter | | |
|---|---|---|---|---|---|---|---|---|---|---|---|---|---|---|---|---|---|---|---|---|---|---|---|---|---|---|---|
| | All | Old | New | All | Old | New | All | Old | New | All | Old | New | All | Old | New | All | Old | New | All | Old | New | All | Old | New | All | Old | New |
| RankStats+ | 13.6 | 20.9 | 4.5 | 12.7 | 28.4 | 5.1 | 12.3 | 27.4 | 5.4 | 15.2 | 33.7 | 4.9 | 16.0 | 34.7 | 5.6 | 17.5 | 38.4 | 4.8 | 18.7 | 40.7 | 4.9 | 16.8 | 36.5 | 5.3 | 22.3 | 48.1 | 4.7 |
| UNO+ | 18.5 | 32.4 | 7.6 | 17.2 | 30.5 | 7.2 | 17.1 | 31.1 | 6.2 | 20.4 | 35.7 | 8.4 | 20.7 | 35.6 | 7.0 | 20.7 | 35.2 | 7.4 | 30.2 | **52.2** | 10.5 | 22.9 | 42.0 | 8.4 | 29.7 | **52.7** | 11.2 |
| ORCA | 21.5 | 23.1 | 19.9 | 21.2 | 23.7 | 18.8 | 21.1 | 23.1 | 19.2 | 20.4 | 22.0 | 18.9 | 20.1 | 22.1 | 18.3 | 22.0 | 25.5 | 18.5 | 19.2 | 20.4 | 18.0 | 22.4 | 20.8 | 19.1 | 24.8 | 31.3 | 18.3 |
| GCD | 23.4 | 22.7 | 20.0 | 22.7 | 20.4 | 31.0 | 21.9 | 20.3 | 19.6 | 25.1 | 25.3 | 21.0 | 23.6 | 22.9 | 20.2 | 23.9 | 23.1 | 20.8 | 29.7 | 31.1 | 24.4 | 27.6 | 26.7 | 24.6 | 35.2 | 36.2 | 30.3 |
| SimGCD | 23.8 | 26.6 | 22.0 | 21.6 | 23.8 | 20.4 | 20.4 | 22.5 | 19.4 | 30.5 | 35.8 | 26.2 | 29.0 | 34.3 | 24.9 | 29.1 | 32.6 | 26.7 | 33.0 | 36.9 | 30.1 | 27.3 | 29.6 | 26.1 | 41.5 | 47.0 | 37.0 |
| HiLo (Ours) | **41.8** | **39.8** | **43.9** | **41.0** | **38.7** | **43.3** | **42.2** | **39.8** | **44.5** | **47.9** | **43.9** | **51.8** | **49.3** | **45.8** | **52.8** | **48.5** | **45.5** | **51.4** | **50.6** | 46.8 | **54.3** | **47.9** | **45.4** | **50.2** | **50.9** | 47.2 | **54.7** |

Table 12: A detailed evaluation of the Scars-C dataset. We assess the performance of each individual corruption.

| Methods | Gaussian Noise | | | Shot Noise | | | Impulse Noise | | | Zoom Blur | | | Snow | | | Frost | | | Fog | | | Speckle | | | Spatter | | |
|---|---|---|---|---|---|---|---|---|---|---|---|---|---|---|---|---|---|---|---|---|---|---|---|---|---|---|---|
| | All | Old | New | All | Old | New | All | Old | New | All | Old | New | All | Old | New | All | Old | New | All | Old | New | All | Old | New | All | Old | New |
| RankStats+ | 8.5 | 16.6 | 1.6 | 8.9 | 16.7 | 1.7 | 7.2 | 13.8 | 1.5 | 11.7 | 22.9 | 0.5 | 8.9 | 17.0 | 1.3 | 11.4 | 21.9 | 0.7 | 16.8 | 32.6 | 1.2 | 12.7 | 24.1 | 1.6 | 17.3 | 34.1 | 1.6 |
| UNO+ | 13.9 | 24.8 | 6.5 | 14.0 | 25.0 | 6.9 | 11.2 | 20.4 | 6.4 | 17.1 | 33.2 | 2.6 | 13.3 | 24.0 | 4.5 | 17.3 | 29.9 | 6.3 | 22.4 | 39.8 | 3.8 | 18.6 | 33.1 | 7.1 | 21.8 | 38.4 | 4.0 |
| ORCA | 12.0 | 31.4 | 9.3 | 13.2 | 31.8 | 9.7 | 11.8 | 29.2 | 9.2 | 14.5 | 38.2 | 7.9 | 12.5 | 32.6 | 9.5 | 15.7 | 36.4 | 10.0 | 20.3 | 47.7 | 5.8 | 17.0 | **39.4** | 10.5 | 21.6 | **48.8** | 10.6 |
| GCD | 17.6 | 24.2 | 10.8 | 17.1 | 24.6 | 11.2 | 14.4 | 20.9 | 11.0 | 23.2 | 31.8 | 8.0 | 18.5 | 25.5 | 8.4 | 23.2 | 31.1 | 10.2 | 27.1 | 40.8 | 5.7 | 22.6 | 30.1 | 12.4 | 31.0 | 43.1 | 7.1 |
| SimGCD | 18.1 | 23.5 | 15.7 | 18.3 | 23.5 | 15.5 | 15.2 | 19.0 | 15.4 | 24.4 | 32.7 | 13.1 | 19.7 | 26.4 | 12.9 | 23.9 | 31.9 | 13.3 | 28.0 | 38.6 | 12.7 | 23.4 | 30.6 | 16.4 | 32.4 | 45.4 | 13.1 |
| HiLo (Ours) | **31.0** | **38.0** | **24.3** | **31.5** | **38.3** | **24.9** | **30.2** | **36.6** | **23.9** | **38.4** | **45.1** | **31.9** | **36.8** | **44.9** | **29.0** | **36.5** | **43.8** | **29.5** | **40.7** | **49.5** | **32.2** | **37.1** | 37.1 | **29.6** | **37.9** | 45.4 | **30.6** |

Table 13: A detailed evaluation of the FGVC-C dataset. We assess the performance of each individual corruption.

| Methods | Gaussian Noise | | | Shot Noise | | | Impulse Noise | | | Zoom Blur | | | Snow | | | Frost | | | Fog | | | Speckle | | | Spatter | | |
|---|---|---|---|---|---|---|---|---|---|---|---|---|---|---|---|---|---|---|---|---|---|---|---|---|---|---|---|
| | All | Old | New | All | Old | New | All | Old | New | All | Old | New | All | Old | New | All | Old | New | All | Old | New | All | Old | New | All | Old | New |
| RankStats+ | 7.3 | 13.6 | 5.0 | 6.3 | 10.7 | 5.8 | 6.0 | 10.7 | 5.3 | 10.1 | 19.9 | 4.3 | 6.2 | 12.5 | 3.8 | 8.9 | 17.7 | 4.1 | 12.5 | 24.4 | 4.5 | 7.6 | 14.0 | 5.2 | 10.5 | 20.1 | 4.9 |
| UNO+ | 15.5 | **25.2** | 5.8 | 13.5 | 22.1 | 4.9 | 13.2 | 20.1 | 6.2 | 20.1 | 28.2 | 12.0 | 15.6 | 21.3 | 9.9 | 17.6 | 25.2 | 9.9 | 19.3 | 26.9 | 11.7 | 16.5 | 27.2 | 5.6 | 20.9 | 29.6 | 12.3 |
| ORCA | 11.9 | 17.3 | 11.1 | 11.1 | 15.6 | 11.3 | 10.9 | 15.9 | 10.3 | 15.2 | 24.3 | 8.5 | 11.3 | 15.4 | 9.1 | 12.6 | 22.1 | 9.3 | 16.7 | 28.9 | 9.1 | 12.2 | 18.8 | 10.4 | 15.0 | 25.1 | 9.3 |
| GCD | 16.0 | 20.1 | 14.3 | 13.8 | 19.1 | 11.5 | 12.3 | 16.0 | 13.4 | 27.7 | 25.4 | 24.1 | 19.1 | 17.7 | 15.2 | 23.9 | 24.0 | 18.2 | 31.8 | 30.1 | 24.7 | 16.1 | 27.0 | 14.9 | 28.7 | 30.7 | 25.9 |
| SimGCD | 16.3 | 16.2 | 18.4 | 14.2 | 14.5 | 16.0 | 13.7 | 13.0 | 16.5 | 28.9 | 31.4 | 28.4 | 20.0 | 22.4 | 19.5 | 24.5 | 29.2 | 21.9 | 31.9 | **37.8** | 28.0 | 16.8 | 18.0 | 17.7 | 29.8 | **32.9** | 28.6 |
| HiLo (Ours) | **28.6** | 25.2 | **32.0** | **26.8** | 24.4 | **29.2** | **27.9** | 24.5 | **31.4** | **36.8** | **34.2** | **39.4** | **27.8** | 27.9 | **27.8** | **33.4** | **30.4** | **36.4** | **35.8** | 34.1 | **37.5** | **30.4** | **30.4** | **32.7** | **33.4** | 32.4 | **34.4** |

# I    ADDITIONAL EXPERIMENTAL RESULTS ON DOMAINNET

As discussed in Section 4.1 and Section 4.2 in the main paper, among the 6 domains in DomainNet, we utilize the 'Real' domain as $\Omega^a$ and each of the other 5 domains serves as $\Omega^b$ in turn. Note that the model is fitted on the partially labelled data from domain $\Omega^a$, which contains labelled and novel classes, and the fully unlabelled data from domain $\Omega^b$. Therefore, though the model does not 'see' the novel classes in both domains $\Omega^a$ and $\Omega^b$, it does 'see' the unlabelled data from both domains, regardless of whether the images are from labelled or novel classes. To more comprehensively measure the model's capability, we further evaluate the performance on the unlabelled images from the remaining 4 domains aside from $\Omega^a$ and $\Omega^b$.

In Table 14, we report the results by considering the 'Infograph' domain as $\Omega^b$. 'Others' denotes the results on the unlabelled data from the remaining 4 domains aside from 'Real' and 'Infograph'. Table 15 shows the evaluation on each domain in 'Others'. Similarly, Table 16 and Table 17 show the results by considering 'Quickdraw' domain as $\Omega^b$; Table 18 and Table 19 show the results by considering 'Sketch' domain as $\Omega^b$; and Table 20 and Table 21 show the results by considering 'Clipart' domain as $\Omega^b$. Our HiLo framework consistently outperforms baseline methods in both 'All' and 'New' performance. Notably, the 'Quickdraw' domain presents greater challenges than other domains due to its highly abstract and difficult-to-recognize images, resulting in unsatisfactory performance for all methods.

Table 14: Evaluation on the DomainNet dataset. The model is trained on the 'Real' and 'Infograph' domains and we report the respective results on 'Real', 'Infograph' and the remaining four domains (*i.e.*, 'Others').

| Methods | Real | | | Infograph | | | Others | | |
|---|---|---|---|---|---|---|---|---|---|
| | All | Old | New | All | Old | New | All | Old | New |
| RankStats+ | 34.2 | 62.4 | 19.6 | 12.5 | **21.9** | 6.3 | 18.5 | **32.1** | 6.4 |
| UNO+ | 42.8 | 69.4 | 29.0 | 10.9 | 15.2 | 8.0 | 18.2 | 28.0 | 9.6 |
| ORCA | 29.1 | 47.7 | 20.1 | 8.6 | 13.7 | 7.1 | 13.8 | 24.8 | 5.4 |
| GCD | 41.9 | 46.1 | 39.0 | 10.9 | 17.1 | 8.8 | 19.0 | 29.1 | 11.1 |
| SimGCD | 52.7 | 67.0 | 44.8 | 11.6 | 15.4 | 9.1 | 20.8 | 28.4 | 14.2 |
| HiLo (Ours) | **64.2** | **78.1** | **57.0** | **13.7** | 16.4 | **11.9** | **23.0** | 28.5 | **18.3** |

Table 15: Evaluation on the DomainNet dataset. Besides the overall performance given in Table 14, we show a detailed performance breakdown for each domain in 'Others'.

| Methods | Painting | | | Quickdraw | | | Sketch | | | Clipart | | |
|---|---|---|---|---|---|---|---|---|---|---|---|---|
| | All | Old | New | All | Old | New | All | Old | New | All | Old | New |
| RankStats+ | 29.6 | **49.2** | 10.0 | 2.5 | 1.6 | **3.4** | 17.4 | **32.2** | 6.5 | 24.4 | **45.5** | 5.8 |
| UNO+ | 30.8 | 44.8 | 16.8 | 2.7 | 2.3 | 3.1 | 17.0 | 27.0 | 9.7 | 22.3 | 37.8 | 8.7 |
| ORCA | 20.0 | 40.2 | 8.1 | 1.6 | 1.8 | 1.2 | 13.2 | 21.1 | 8.0 | 20.5 | 36.0 | 4.1 |
| GCD | 30.8 | 45.1 | 18.4 | **3.6** | **4.7** | 2.5 | 18.8 | 26.4 | 11.2 | 22.9 | 40.0 | 12.3 |
| SimGCD | 35.9 | 45.6 | 26.3 | 2.1 | 1.7 | 2.5 | 20.8 | 29.3 | 14.5 | 24.5 | 36.9 | 13.6 |
| HiLo (Ours) | **40.1** | 46.1 | **35.8** | 2.0 | 2.2 | 1.5 | **22.6** | 29.4 | **17.6** | **26.6** | 36.3 | **18.1** |

Table 16: Evaluation on the DomainNet dataset. The model is trained on the 'Real' and 'Quickdraw' domains and we report the respective results on 'Real', 'Quickdraw' and the remaining four domains (*i.e.*, 'Others').

| Methods | Real | | | Quickdraw | | | Others | | |
|---|---|---|---|---|---|---|---|---|---|
| | All | Old | New | All | Old | New | All | Old | New |
| RankStats+ | 34.1 | 62.5 | 19.5 | 4.1 | 4.4 | 3.9 | 21.0 | **37.4** | 7.2 |
| UNO+ | 31.1 | 60.0 | 16.1 | 6.3 | 5.8 | 6.8 | 18.6 | 32.2 | 7.0 |
| ORCA | 19.2 | 39.1 | 15.3 | 3.4 | 3.5 | 3.2 | 15.6 | 28.4 | 8.1 |
| GCD | 37.6 | 41.0 | 35.2 | 5.7 | 4.2 | 6.9 | 21.9 | 34.3 | 12.2 |
| SimGCD | 47.4 | 64.5 | 37.4 | 6.6 | 5.8 | 7.5 | 22.9 | 33.8 | 13.8 |
| HiLo (Ours) | **58.6** | **76.4** | **52.5** | **7.4** | **6.9** | **8.0** | **25.9** | 32.5 | **20.4** |

Table 17: Evaluation on the DomainNet dataset. Besides the overall performance given in Table 16, we show a detailed performance breakdown for each domain in 'Others'.

| Methods | Painting | | | Sketch | | | Clipart | | | Infograph | | |
|---|---|---|---|---|---|---|---|---|---|---|---|---|
| | All | Old | New | All | Old | New | All | Old | New | All | Old | New |
| RankStats+ | 29.6 | **49.0** | 10.2 | 17.1 | **32.1** | 6.1 | 24.8 | **45.4** | 6.7 | 12.6 | **23.1** | 5.7 |
| UNO+ | 26.8 | 43.7 | 9.9 | 14.7 | 25.6 | 6.6 | 20.7 | 38.4 | 5.1 | 12.2 | 21.0 | 6.4 |
| ORCA | 22.2 | 40.9 | 10.1 | 11.9 | 22.4 | 7.1 | 17.5 | 35.6 | 5.7 | 10.3 | 18.7 | 6.6 |
| GCD | 32.9 | 45.7 | 21.4 | 18.5 | 30.5 | 10.8 | 23.5 | 39.0 | 10.7 | 13.8 | 22.1 | 7.6 |
| SimGCD | 33.8 | 45.1 | 22.5 | 19.4 | 30.1 | 11.5 | 24.0 | 38.5 | 11.4 | 14.5 | 21.6 | 9.8 |
| HiLo (Ours) | **38.6** | 45.1 | **32.2** | **22.9** | 28.8 | **18.5** | **26.0** | 36.4 | **16.9** | **16.2** | 19.8 | **13.9** |

Table 18: Evaluation on the DomainNet dataset. The model is trained on the 'Real' and 'Sketch' domains and we report the respective results on 'Real', 'Sketch' and the remaining four domains (*i.e.*, 'Others').

| Methods | Real | | | Sketch | | | Others | | |
|---|---|---|---|---|---|---|---|---|---|
| | All | Old | New | All | Old | New | All | Old | New |
| RankStats+ | 34.2 | 62.0 | 19.8 | 17.1 | **31.1** | 6.8 | 17.3 | **30.0** | 6.1 |
| UNO+ | 43.7 | 72.5 | 28.9 | 12.5 | 17.0 | 9.2 | 17.4 | 26.4 | 9.5 |
| ORCA | 32.5 | 50.0 | 23.9 | 11.4 | 14.5 | 7.2 | 13.3 | 23.1 | 9.1 |
| GCD | 48.0 | 53.8 | 45.3 | 16.6 | 22.4 | 11.1 | 20.7 | 25.8 | 15.8 |
| SimGCD | 62.4 | 77.6 | 54.6 | 16.4 | 20.2 | 13.6 | 20.4 | 25.4 | 16.1 |
| HiLo (Ours) | **63.3** | **77.9** | **55.9** | **19.4** | 22.4 | **17.1** | **21.3** | 25.8 | **17.4** |

Table 19: Evaluation on the DomainNet dataset. Besides the overall performance given in Table 18, we show a detailed performance breakdown for each domain in 'Others'.

| Methods | Painting | | | Quickdraw | | | Clipart | | | Infograph | | |
|---|---|---|---|---|---|---|---|---|---|---|---|---|
| | All | Old | New | All | Old | New | All | Old | New | All | Old | New |
| RankStats+ | 29.7 | **49.2** | 10.2 | 2.3 | 2.1 | 2.4 | 24.6 | **45.9** | 5.9 | 12.5 | **22.6** | 5.9 |
| UNO+ | 30.8 | 44.0 | 17.6 | 2.4 | 2.4 | 2.3 | 23.1 | 38.0 | 10.1 | 13.2 | 21.2 | 7.9 |
| ORCA | 23.1 | 39.1 | 17.2 | **2.5** | **3.0** | 2.0 | 19.7 | 33.1 | 10.0 | 8.9 | 18.1 | 7.0 |
| GCD | 32.6 | 40.1 | 31.5 | 1.6 | 1.9 | 1.5 | 24.1 | 31.1 | 14.9 | 14.1 | 16.2 | 10.2 |
| SimGCD | 38.7 | 44.7 | 32.7 | 1.9 | 1.2 | **2.5** | 25.2 | 35.3 | 16.3 | 15.8 | 20.3 | 12.8 |
| HiLo (Ours) | **39.8** | 44.7 | **34.9** | 1.9 | 2.0 | 1.7 | **27.2** | 35.9 | **19.6** | **16.2** | 20.5 | **13.4** |

Table 20: Evaluation on the DomainNet dataset. The model is trained on the 'Real' and 'Clipart' domains and we report the respective results on 'Real', 'Clipart' and the remaining four domains (*i.e.*, 'Others').

| Methods | Real | | | Clipart | | | Others | | |
|---|---|---|---|---|---|---|---|---|---|
| | All | Old | New | All | Old | New | All | Old | New |
| RankStats+ | 34.0 | 62.4 | 19.4 | 24.1 | **45.1** | 6.2 | 15.8 | **27.0** | 6.4 |
| UNO+ | 44.5 | 66.1 | 33.3 | 21.9 | 35.6 | 10.1 | 16.2 | 23.2 | 10.5 |
| ORCA | 32.0 | 49.7 | 23.9 | 19.1 | 31.8 | 4.3 | 13.7 | 19.9 | 8.6 |
| GCD | 47.7 | 53.8 | 44.3 | 22.4 | 34.4 | 16.0 | 18.0 | 24.1 | 12.1 |
| SimGCD | 61.6 | 77.2 | 53.6 | 23.9 | 31.5 | 17.3 | 19.2 | 23.6 | 15.6 |
| HiLo (Ours) | **63.8** | **77.6** | **56.6** | **27.7** | 34.6 | **21.7** | **19.8** | 23.6 | **16.8** |

Table 21: Evaluation on the DomainNet dataset. Besides the overall performance given in Table 20, we show a detailed performance breakdown for each domain in 'Others'.

| Methods | Painting | | | Quickdraw | | | Sketch | | | Infograph | | |
|---|---|---|---|---|---|---|---|---|---|---|---|---|
| | All | Old | New | All | Old | New | All | Old | New | All | Old | New |
| RankStats+ | 30.0 | **50.3** | 9.7 | 2.6 | 2.3 | 2.9 | 17.4 | **31.9** | 6.8 | 13.1 | **23.6** | 6.2 |
| UNO+ | 31.5 | 43.3 | 19.6 | 2.8 | 2.1 | **3.6** | 17.3 | 26.8 | 10.2 | 13.3 | 20.6 | 8.5 |
| ORCA | 29.3 | 36.9 | 9.2 | 1.3 | 1.5 | 1.2 | 13.7 | 21.9 | 8.3 | 10.3 | 19.4 | 6.3 |
| GCD | 33.4 | 40.4 | 22.2 | **3.6** | **5.7** | 2.2 | 19.5 | 27.7 | 12.7 | 15.5 | 22.7 | 11.1 |
| SimGCD | 39.0 | 45.9 | 32.1 | 0.8 | 0.5 | 1.1 | 21.1 | 27.3 | 16.5 | 15.9 | 20.8 | 12.7 |
| HiLo (Ours) | **40.7** | 46.3 | **35.1** | 1.3 | 0.4 | 2.3 | **21.2** | 26.9 | **17.0** | **15.9** | 20.6 | **12.8** |

## J    HiLo on the vanilla GCD setting

Although not explicitly designed for the vanilla GCD, we evaluate HiLo's effectiveness without domain shifts. We perform experiments on ImageNet-100 and SSB. Our HiLo framework outperforms the state-of-the-art GCD method, as indicated in Table 22. We hypothesize that subtle covariate shifts may still be present within the same distribution (*e.g.*, varying 'Real' backgrounds with identical semantics), which can still be handled by HiLo effectively.

Table 22: Evaluation of HiLo on ImageNet-100 and SSB under the vanilla GCD setting. HiLo achieves better results than the SoTA GCD method.

| Method | ImageNet-100 | | | SSB | | |
|---|---|---|---|---|---|---|
| | All | Old | New | All | Old | New |
| SimGCD | 83.0 | 93.1 | 77.9 | 56.1 | 65.5 | 51.5 |
| HiLo (Ours) | **83.4** | **93.5** | **78.1** | **59.2** | **66.2** | **54.9** |

# K  EFFECTS OF DIFFERENT OUTPUT DIMENSIONS FOR THE SEMANTIC AND DOMAIN HEADS

In the main paper, we assume access to the ground-truth values of both the semantic class and domain (*i.e.,* $k_s$ and $k_d$). However, in real-world scenarios, these values are often unknown. Therefore, it is essential to assess the stability of our model's performance when assigning guesses to the varying output quantities of semantic class and domain type.

We employ different output dimensions for the semantic head and domain head. For the semantic head, we experiment with $k_s \in \{200, 1000, 2000, 5000, 10000\}$ using all 5 severity levels. For the domain head, we experiment with $k_d \in \{2, 10, 20, 50, 100\}$ and corruptions with the highest severity level. Table 23 reports the accuracy with different $k_s$ and $k_d$ values, with the optimal number utilized to fix one output size while exploring the other. The highest performance is achieved when $k_s = |\mathcal{Y}^l \cup \mathcal{Y}^u|$ and $k_d = 10$. Performance declines with increasing $k_s$ or $k_d$. As it is tractable to roughly estimate the number of domains the model may handle, our method's insensitivity to the domain axis output size selection.

Table 23: Sensitivity analysis of the output size on CUB-C dataset. The inappropriate selection of $k_s$ and $k_d$ would predispose to poor performance for the semantic head while the domain head is relatively robust to the output size.

| | Sem. Head | | | | | | | Dom. Head | | | | | |
| | Original | | | Corrupted | | | | Original | | | Corrupted | | |
| Size | All | Old | New | All | Old | New | Size | All | Old | New | All | Old | New |
|---|---|---|---|---|---|---|---|---|---|---|---|---|---|
| $k_s = 200$ | 56.8 | 54.0 | 60.3 | 52.0 | 53.6 | 50.5 | $k_d = 2$ | 43.5 | 45.8 | 40.2 | 35.1 | 37.4 | 32.9 |
| $k_s = 1000$ | 47.5 | 51.0 | 35.1 | 38.1 | 48.3 | 30.9 | $k_d = 10$ | 44.2 | 46.2 | 43.0 | 36.3 | 39.1 | 34.7 |
| $k_s = 2000$ | 40.0 | 48.7 | 30.1 | 31.4 | 40.0 | 25.1 | $k_d = 20$ | 43.0 | 44.6 | 40.0 | 34.9 | 36.5 | 32.3 |
| $k_s = 5000$ | 30.7 | 43.1 | 22.2 | 23.8 | 28.1 | 21.8 | $k_d = 500$ | 37.5 | 41.4 | 34.9 | 30.3 | 33.1 | 28.7 |
| $k_s = 10000$ | 14.2 | 30.1 | 10.0 | 12.1 | 13.9 | 13.1 | $k_d = 1000$ | 35.0 | 38.3 | 33.2 | 28.9 | 31.5 | 27.3 |

## L    UNKNOWN CATEGORY NUMBER

As the total number of semantic categories cannot be accessed in the real-world setting, we evaluate our HiLo with an estimated number of categories using an off-the-shelf method Vaze et al. (2022) on CUB (see Table 24). We find that our method consistently outperforms the strong baseline when the exact number of categories is unknown.

Table 24: Performance of HiLo and the baseline method SimGCD with an estimated number of categories on CUB. Bold values represent the best results. 'GT' denotes the ground truth; 'Est.' denotes the estimation.

| Method | $|\mathcal{C}|$ | Original | | | Corrupted | | |
|---|---|---|---|---|---|---|---|
| | | All | Old | New | All | Old | New |
| SimGCD Wen et al. (2023) | GT (200) | 31.9 | 33.9 | 29.0 | 28.8 | 31.6 | 25.0 |
| **HiLo (Ours)** | GT (200) | 56.8 | 54.0 | 60.3 | 52.0 | 53.6 | 50.5 |
| SimGCD Wen et al. (2023) | Est. (257) | 29.5 | 32.4 | 28.0 | 27.6 | 29.7 | 24.1 |
| **HiLo (Ours)** | Est. (257) | 55.9 | 52.9 | 59.2 | 51.2 | 52.8 | 49.5 |

# M  HYPERPARAMETER CHOICES FOR HILO COMPONENTS

The hyperparameters of HiLo can be grouped via each component: (a) PatchMix (*i.e.*, $\beta_k$); (b) representation learning and parametric classification losses (*i.e.*, $\tau$, $\lambda$, $\epsilon$); (c) curriculum learning (*i.e.*, $r_0$, $r'$ and $t'$). We follow Zhu et al. (2023); Wen et al. (2023) to choose values for the shared hyperparameters in (a) and (b) respectively.

As summarized in Table 25, we choose the hyperparameters in (a) and (b) following Zhu et al. (2023) and Wen et al. (2023) respectively. For the hyperparameters in (c), we choose the values through the validation split of the labelled data in the 'Orignal' domain.

Table 25: Hyperparameter choices for HiLo components.

| Hyperparameters | Value | Descriptions |
|---|---|---|
| $\tau_u$ | 0.07 | Suggested values following Wen et al. (2023) |
| $\tau_c$ | 1.0 | Suggested values following Wen et al. (2023) |
| $\tau_s$ | 0.1 | Suggested values following Wen et al. (2023) |
| $\tau_t$ | 0.07 | Suggested values following Wen et al. (2023) |
| $\lambda$ | 0.35 | Suggested values following Wen et al. (2023) |
| $\beta$ | $\sim \text{Beta}(\log(1+e), \log(1+e))$ | Suggested value following Zhu et al. (2023) |
| $\varepsilon$ | 0.1 | Choose through the validation split of the labelled data in the 'Orignal' domain (see Figure 7(a)) |
| $r'$ | 0.05 | Choose through the validation split of the labelled data in the 'Orignal' domain (see Figure 7(b)) |
| $r_0$ | 0 | Choose through the validation split of the labelled data in the 'Orignal' domain (see Figure 7(c)) |
| $t'$ | 80 | Choose through the validation split of the labelled data in the 'Orignal' domain (see Figure 7(d)) |

In Figure 7, we report results on CUB-C with varying values of $\varepsilon, r_0, r', t'$ that are specific to HiLo. We find that the order of samples (determined by $r_0, r', t'$) with different difficulties has a great influence on performance on both the source domain and target domains.

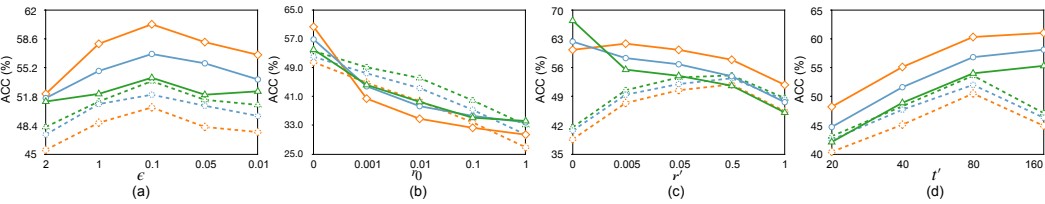

Figure 7: The impact of varying values of $\varepsilon$, $r_0$, $r'$ and $t'$ investigated on the CUB-C dataset. Hyperparameters for curriculum sampling (*i.e.*, $r_0, r', t'$) have a great influence on performance on both source domain and target domains.

# N    EFFECTS OF LEARNING RATES

As learning rate is a key hyperparameter for all methods, we present results using different learning rates for our method and the baselines on the CUB-C datasets. We experiment with three different learning rates, 0.1, 0.05, and 0.01, for all the methods, using the SGD optimizer with the suggested weight decay and momentum in the original papers. 0.1 appears to be the best choice among the three values for RankStats+, UNO+, and SimGCD, while 0.05 is a better choice for our method. Among the compared methods, we can see that the performance variation is relatively large for GCD and SimGCD among these three values. The variation is relatively small for RankStat+, UNO+, and ORCA, while their performance is notably inferior to GCD and SimGCD. In contrast, our method has a very small performance variation while significantly outperforms all other methods.

Table 26: Performance comparison on CUB-C with three different learning rates.

| Method | Learning Rate | Original | | | Corrupted | | |
|---|---|---|---|---|---|---|---|
| | | All | Old | New | All | Old | New |
| RankStat+ | 0.1 | 19.3 | 22.0 | 15.4 | 13.6 | 23.9 | 4.5 |
| | 0.05 | 17.1 | 24.9 | 12.7 | 11.9 | 16.7 | 8.5 |
| | 0.01 | 15.0 | 17.1 | 10.7 | 9.1 | 15.5 | 3.8 |
| UNO+ | 0.1 | 25.9 | 40.1 | 21.3 | 21.5 | 33.4 | 8.6 |
| | 0.05 | 23.8 | 37.2 | 18.8 | 20.2 | 34.0 | 7.1 |
| | 0.01 | 22.8 | 35.7 | 17.9 | 19.5 | 33.2 | 5.8 |
| ORCA | 0.1 | 17.3 | 22.6 | 13.8 | 20.9 | 22.6 | 17.4 |
| | 0.05 | 18.2 | 22.8 | 14.5 | 21.5 | 23.1 | 18.9 |
| | 0.01 | 17.4 | 22.1 | 13.2 | 20.8 | 23.6 | 15.8 |
| GCD | 0.1 | 26.6 | 27.5 | 25.7 | 25.1 | 28.7 | 22.0 |
| | 0.05 | 24.7 | 25.4 | 23.8 | 24.0 | 28.2 | 20.8 |
| | 0.01 | 48.1 | 53.1 | 47.0 | 33.1 | 37.2 | 29.9 |
| SimGCD | 0.1 | 31.9 | 33.9 | 29.0 | 28.8 | 31.6 | 25.0 |
| | 0.05 | 29.2 | 30.7 | 27.1 | 25.0 | 26.5 | 24.0 |
| | 0.01 | 26.3 | 27.0 | 25.9 | 21.8 | 21.4 | 23.5 |
| Ours | 0.1 | 56.0 | 54.1 | 58.9 | 50.8 | 52.4 | 48.1 |
| | 0.05 | 56.8 | 54.0 | 60.3 | 52.0 | 53.6 | 50.5 |
| | 0.01 | 54.7 | 60.1 | 56.4 | 48.1 | 49.0 | 47.6 |

# O    STABILITY OF DIFFERENT METHODS

As the differences in the results of the GCD benchmark tests can be very large, we obtain the averaged results in Table 2 and Table 3 by conducting three independent runs for each method on both DomainNet and SSB-C. Here we visualize the bar chart of 'All' classes ACC for each methods and list the corresponding error lines generated by the these independent runs. We notice that the error bars of ORCA and SimGCD exhibit significant oscillations.

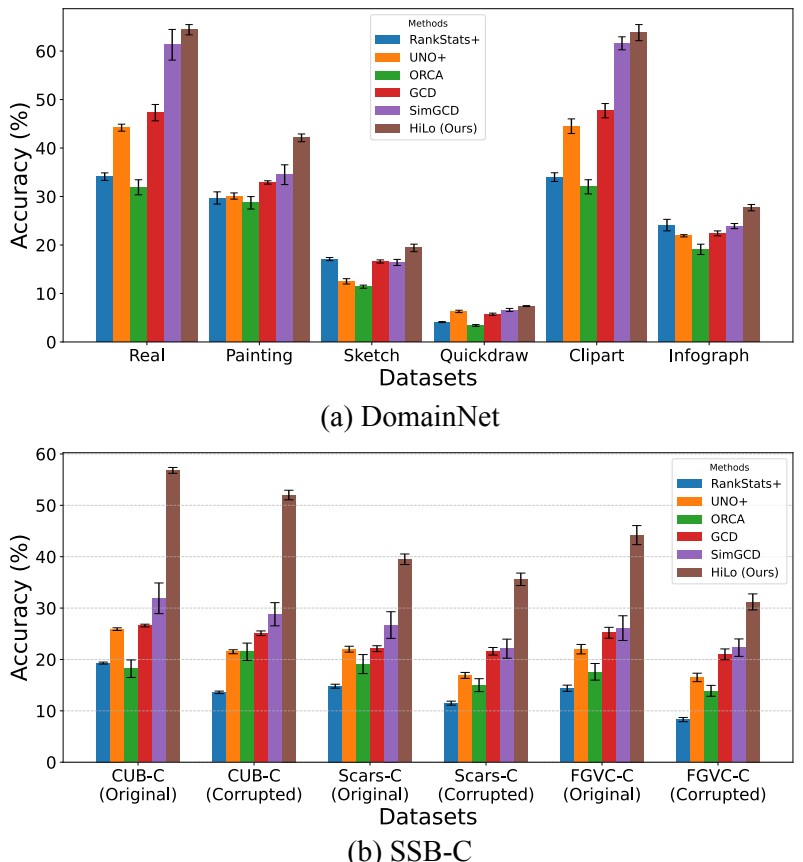

(a) DomainNet

(b) SSB-C

Figure 8: The 'All' ACC results are averaged by conducting three independent runs for each method on both DomainNet and SSB-C.

## P    QUALITATIVE RESULTS

We provide qualitative results on DomainNet and CUB-C. In Figure 9, we present the visualization by first applying PCA to the domain features and semantic features obtained through $\tilde{\mathcal{H}}$, and then plotting the corresponding images. As can be seen, the images are naturally clustered according to their domains and semantics, demonstrating that HiLo successfully learns domain-specific and semantic-specific features.

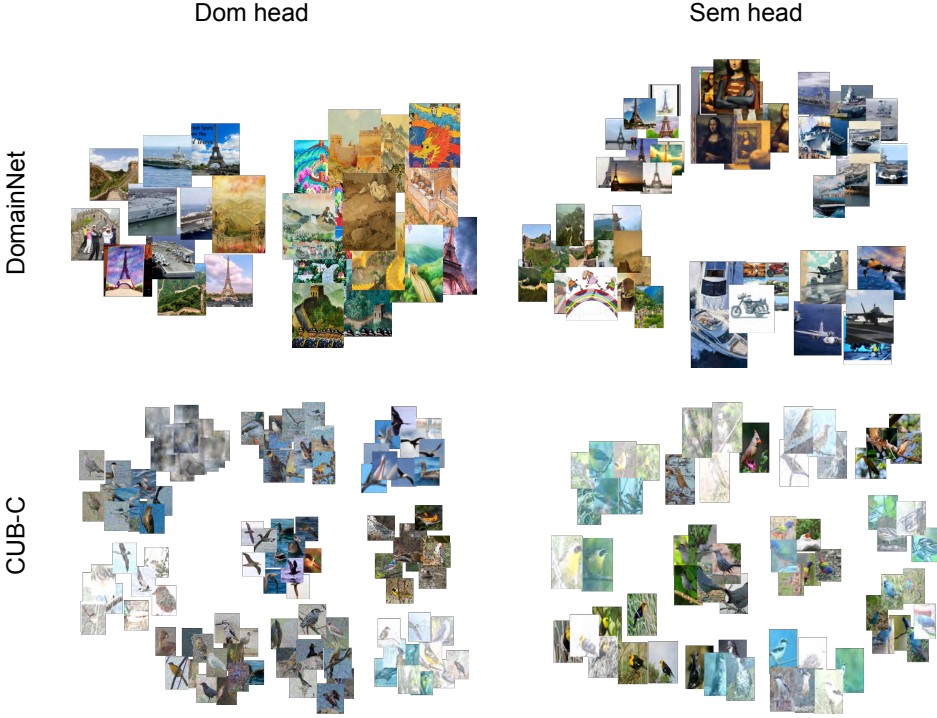

Figure 9: Visualization of domain and semantic features via projecting them through PCA. We randomly sample instances from the entire dataset and apply PCA to project the semantic and domain features into a 2-dimensional space. The domain branch tends to cluster images based on covariate features, while the semantic branch clusters images based on categories. Best viewed in PDF with zoom.

The attention map offers valuable insights into the focus of Transformer-based models on the input. We obtain the attention maps for the `CLS` token from multiple attention heads in the final layer of the ViT backbone, highlighting the top 10% most attended patches in Figure 10. We observe that, compared with the baseline, HiLo is much more effective in focusing on the foreground object even in the presence of significant domain shifts (*e.g.*, painting style, foggy weather). This demonstrates that HiLo is robust to domain shifts and remains unaffected by potential spurious correlations between semantic features and low-level statistics.

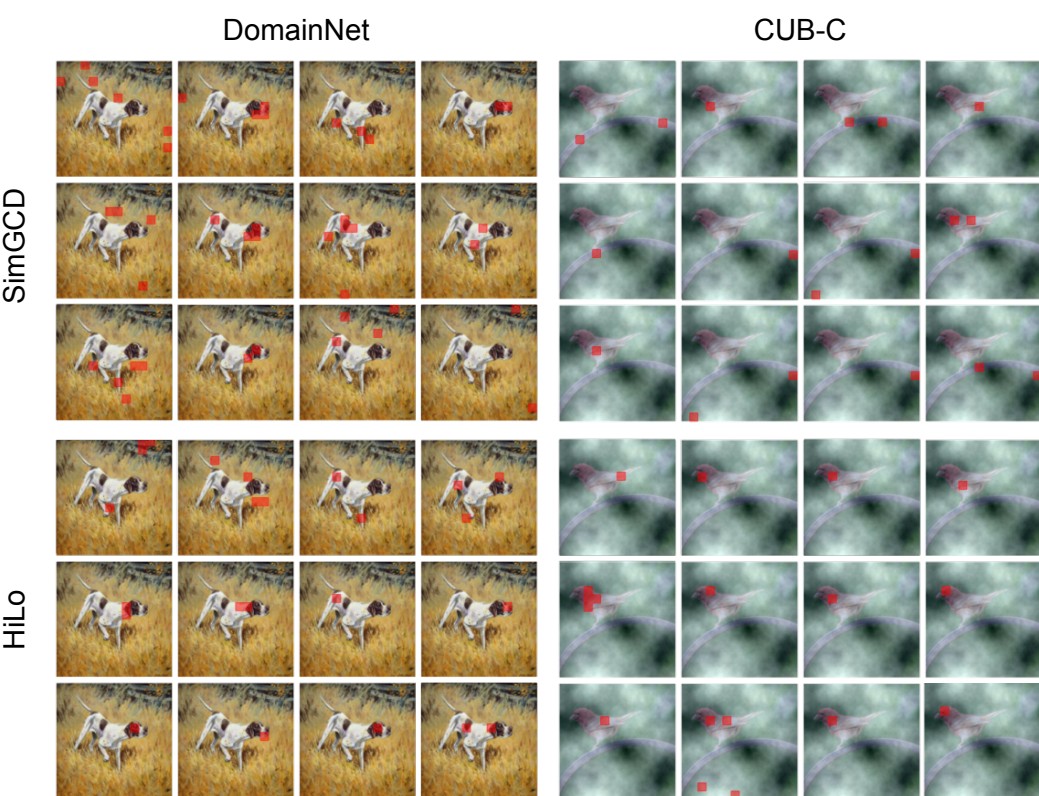

Figure 10: Visualization of the attention map for different heads in the last layer on DomainNet and CUB-C. We highlight the attended regions with top 10% contribution in red. Compared with SimGCD, the attention maps of HiLo consistently focus on the foreground object without affecting by the strong domain shifts of painting style and foggy weather.

## Q    BROADER IMPACTS AND LIMITATIONS

Our study aims to extend AI systems' capabilities from closed-world to open-world scenarios, particularly enhancing next-generation AI systems to categorize and organize open-world data autonomously. Despite promising results on public datasets, our method has limitations. First, interpretability needs improvement, as the underlying decision-making principles remain unclear. Second, cross-domain robustness is inadequate. Although our method has achieved the best overall and new class discovery results in the GCD setting with domain shifts, performance still has significant room for improvement. Third, the novel domains we investigated in the paper are still limited. Domain and class imbalance present additional challenges in GCD scenarios. Our current method was not specifically developed to handle these issues, which are important areas for future work.

