# OpenReview forum: "HiLo: A Learning Framework for Generalized Category Discovery Robust to Domain Shifts"
_ICLR.cc/2025/Conference — ICLR 2025 Poster_

### Official Review · Reviewer_jy36 · 2024-10-28

**Soundness:** 2
**Presentation:** 3
**Contribution:** 2
**Rating:** 6
**Confidence:** 4

**Summary:**

This paper introduces a new problem setting: Generalized Category Discovery (GCD) with domain shift. The authors leverage techniques from domain adaptation and curriculum learning to propose a new method called HiLo. Comprehensive experiments on the proposed benchmark demonstrate substantial improvements.

**Strengths:**

The paper proposes a new problem setting and proposes a HiLo, which combines multiple techniques from domain adaption and achieves better results.

**Weaknesses:**

1. The novelty of the method appears limited, as it seems to combine various techniques from different domains.

2. The comparison with UniOT should be included in the main results. Since the proposed setting is similar to universal domain adaptation, it is essential to compare methods from both domains in the main results.

Minor：

Missing citation for the following important paper

[1] Rastegar et al. Learn to Categorize or Categorize to Learn? Self-Coding for Generalized Category Discovery. NeurIPS 2023.

[2] Gu et al. Class-relation Knowledge Distillation for Novel Class Discovery. ICCV2023.

**Questions:**

Please clarify the novelty of the proposed method, and include more comparisons with UinOT in the main results.

---

> ### Author Response · Authors · 2024-11-25
>
> > The novelty of the method appears limited, as it seems to combine various techniques from different domains.
>
> Please see the General Response for our response regarding novelty of the method.
>
>
> > The comparison with UniOT should be included in the main results. Since the proposed setting is similar to universal domain adaptation, it is essential to compare methods from both domains in the main results.
>
> Following the suggestion, we have conducted experiments using UniOT on SSB-C. The results are presented in Table S2 and also incorporated into Table 3 in the main paper. Our HiLo continues to outperform all other methods.
>
> Table S2: Evaluation on SSB-C. Bold values represent the best results.
> | | CUB-C Original |  |  | CUB-C Corrupted |  |  | Scars-C Original |  |  | Scars-C Corrupted |  |  | FGVC-C Original |  |  | FGVC-C Corrupted |  |  |
> |-----------------|-----|-----|-----|-----|-----|-----|-----|-----|-----|-----|-----|-----|-----|-----|-----|-----|-----|-----|
> | | All | Old | New | All | Old | New | All | Old | New | All | Old | New | All | Old | New | All | Old | New |
> | SimGCD | 31.9 | 33.9 | 29.0 | 28.8 | 31.6 | 25.0 | 26.7 | 39.6 | 25.6 | 22.1 | 30.5 | 14.1 | 26.1 | 28.9 | 25.1 | 22.3 | 23.2 | 21.4 |
> | UniOT | 27.5 | 29.3 | 26.8 | 27.3 | 33.2 | 22.5 | 24.3 | 37.5 | 22.3 | 22.9 | 31.4 | 13.7 | 27.3 | 29.8 | 22.5 | 21.6 | 23.5 | 19.6 |
> | HiLo (Ours) | **56.8** | **54.0** | **60.3** | **52.0** | **53.6** | **50.5** | **39.5** | **44.8** | **37.0** | **35.6** | **42.9** | **28.4** | **44.2** | **50.6** | **47.4** | **31.2** | **29.0** | **33.4** |
>
>
>
> > Missing citation for the following important paper
>
> We have included these two works in our related work following the suggestion.

---

> > ### Comment · Reviewer_jy36 · 2024-11-27
> >
> > Thank authors' detailed feedback. My concerns are resolved. Therefore, I maintain the initial score.

---

> > > ### Author Response · Authors · 2024-12-01
> > >
> > > Dear Reviewer jy36,
> > >
> > > We are pleased that our responses have addressed your concerns. Thank you very much for your insightful suggestions and valuable efforts, which are crucial for enhancing the quality of our paper.

---

### Official Review · Reviewer_L4eA · 2024-11-02

**Soundness:** 2
**Presentation:** 2
**Contribution:** 2
**Rating:** 6
**Confidence:** 3

**Summary:**

The paper introduces the HiLo framework, a learning method aimed at tackling Generalized Category Discovery (GCD) under domain shifts. HiLo addresses challenges in categorizing both seen and unseen categories across distinct domains within partially labeled datasets, leveraging a multi-faceted approach: mutual information minimization to separate domain and semantic features, PatchMix for augmented domain adaptation, and a curriculum learning strategy. The proposed method is evaluated on synthetic and real-world domain-shift datasets, showing substantial improvements over existing GCD models.

**Strengths:**

1. The paper presents an innovative GCD approach by combining mutual information minimization with domain-specific data augmentation and curriculum learning to handle domain shifts effectively.
2. Extensive evaluation on both synthetic (SSB-C) and real-world (DomainNet) benchmarks demonstrates the model's robustness and its superiority over baseline GCD models, especially under domain-shifted conditions.

**Weaknesses:**

1. In the "Problem statement," the following sentence is unclear: "The objective of GCD is ... with singleton cardinalities for the latter." The author needs to differentiate between the GCD task setting and the domain shift GCD task setting, so this statement should be revised for clarity and precision.
2. The font sizes of the tables are not standardized, and the font in table 2 is too small.
3. I'm curious as to how many runs each of the authors' experimental results were derived from, and given that the differences in the results of the GCD benchmark tests can be very large, the authors should have listed the error lines generated by the three independent runs.

**Questions:**

1. For the experimental results of the ORCA method, what is the backbone used by the authors?
2. Were any curriculum learning alternatives considered, such as adaptive weighting based on difficulty or dynamic sample weighting? A brief discussion on these choices would clarify why the current approach was favored.

---

> ### Author Response · Authors · 2024-11-25
>
> > The author needs to differentiate between the GCD task setting and the domain shift GCD task setting, so this statement should be revised for clarity and precision.
>
>
> Following the suggestion, we have revised the corresponding sentence in Line 164-167 to “The objective of GCD with domain shifts is to classify all unlabelled images in $\mathcal{D}^u$ (from either $\Omega^a$ or $\Omega^b$) using only the labels in $\mathcal{D}^l$. This is different from the setting of NCD with domain shift and GCD, which assumes $\mathcal{Y}^l\cap\mathcal{Y}^u=\emptyset$ for the former and $\Omega^a=\Omega^b$ with $|\Omega^a|=|\Omega^b|=1$ for the latter.”
>
>
> > The font sizes of the tables are not standardized, and the font in table 2 is too small.
>
>
> Thanks for the suggestion. We will properly fix it in our final version. Specifically, we will report the average results across different domains in the main paper and move the breakdown evaluation of each domain shift to Appendix.
>
>
> > The authors should have listed the error lines generated by the three independent runs.
>
>
> All results in Table 2 and 3 are averaged by three trials with different random seeds. Following your suggestion, we have added the bar chars with corresponding error bars for different methods across DomainNet and SSB-C in Appendix O.
>
>
> > For the experimental results of the ORCA method, what is the backbone used by the authors?
>
>
> We use the pretrained DINO model as the backbone for all methods to ensure fair comparison, which is a common practice in the GCD literature (Wen et al. (2023); Wang et al. (2024)).
>
>
> > Were any curriculum learning alternatives considered, such as adaptive weighting based on difficulty or dynamic sample weighting? A brief discussion on these choices would clarify why the current approach was favored.
>
> Indeed, there are several common curriculum learning strategies: (1) difficulty-based adaptive weighting, which adjusts sample weights based on model performance [S9]; (2) dynamic sample weighting, which updates weights during training based on learning progress [S10]. Our method aligns with these approaches in principle, as domain shift naturally correlates with learning difficulty - samples with larger shifts from the source domain are inherently more challenging to learn from. However, rather than computing difficulty measures during training, we pre-compute sampling weights based on domain shifts before training begins. This design choice offers two key benefits: (1) computational efficiency by avoiding per-iteration difficulty assessment, and (2) more stable training by preventing potential oscillations in difficulty estimates that can occur with dynamic weighting. This approach captures similar intuitions about progressive learning.
>
>
> To address this comment, we further explored one of each type, CL [S8] and Self-Paced Learning (SPL) [S9]. For CL, we implemented it by gradually including more difficult samples based on classification loss, starting with 20% of the easiest samples and increasing by 10% every 20 epochs. We also integrated SPL into our framework by adding a weighted loss term $\lambda||v||1 + \sum{i}v_i\ell_i$, where $v_i$ indicates sample weights and $\ell_i$ is the classification loss for sample $i$. While both CL and SPL improve over the baseline SimGCD (60.1/33.2 on Real/Painting domains), achieving 62.0/34.7 and 62.8/35.0 respectively for 'All' classes, they still fall notably short of our method's performance (64.4/42.1 as shown in Table S1 and Table 6 in the main paper). In contrast, our pre-computed domain shift-based sampling achieves better results, demonstrating the effectiveness of using domain shifts as a proxy for learning difficulty.
>
>
> Table S1: Evaluation on DomainNet. Bold values represent the best results.
> | | Real |  |  | Painting |  |  |
> |-----------------|-----|-----|-----|-----|-----|-----|
> | Methods | All | Old | New | All | Old | New |
> | SimGCD | 61.3 | **77.8** | 52.9 | 34.5 | 35.6 | 33.5 |
> | HiLo | **64.4** | 77.6 | **57.5** | **42.1** | **42.9** | **41.3** |
> | HiLo + CL - curriculum sampling | 62.0 | 75.9 | 53.2 | 34.7 | 35.8 | 33.8 |
> | HiLo + SPL - curriculum sampling | 62.8 | 76.5 | 54.5 | 35.0 | 36.1 | 34.0 |
>
>
> [S8] Bengio, Yoshua, et al. "Curriculum learning." ICML. 2009.
>
>
> [S9] Kumar, M., Benjamin Packer, and Daphne Koller. "Self-paced learning for latent variable models." NeurIPS. 2010.

---

> > ### Comment · Reviewer_L4eA · 2024-11-28
> >
> > I appreciate the authors' response, and I will maintain the initial score.

---

> > > ### Author Response · Authors · 2024-12-01
> > >
> > > Dear Reviewer L4eA,
> > >
> > > Thanks very much for your time and valuable comments. We appreciate your positive feedback.

---

### Official Review · Reviewer_RNZQ · 2024-11-04

**Soundness:** 2
**Presentation:** 3
**Contribution:** 3
**Rating:** 8
**Confidence:** 3

**Summary:**

Generalized Category Discovery (GCD) is a challenging task where, given a partially labeled dataset, the model must classify all unlabeled instances. This paper introduces a new task and method to handle the GCD problem when the unlabeled data contains images from different domains. In terms of the method, the HiLo architecture and learning framework involves extracting "low-level" (early layers) and "high-level" (late layers) features from a vision Transformer and decoupling domain and semantic features by minimizing the mutual information between the two sets of features. The PatchMix contrastive learning method is introduced into the GCD task, with its objective function extended to enable the utilization of both labeled and unlabeled data for training. Curriculum learning is adopted, gradually increasing the sampling probability weight of samples predicted to be from unknown domains to enhance the model's robustness to domain shifts. Experiments are conducted on the DomainNet and the SSB-C benchmark datasets constructed based on the Semantic Shift Benchmark (SSB). The experimental results show that HiLo significantly outperforms existing category discovery models, validating the effectiveness of the method.

**Strengths:**

1. The HiLo architecture extracts features from different layers of the vision Transformer and decouples domain and semantic features by minimizing mutual information. This feature processing method, based on the neural network hierarchical structure and information theory, provides a more effective feature representation for category discovery in the presence of domain shifts and avoids the problem of feature confusion in traditional methods.

2. The PatchMix method is introduced into the GCD task and innovatively extended. By adjusting its objective function, it can adaptively utilize labeled and unlabeled data for training. This extension not only combines the advantages of data augmentation but also flexibly adjusts the learning process according to the nature of different data, enhancing the model's ability to learn data from different domains and categories.

3. The curriculum learning method is employed, which dynamically adjusts the sampling probability weights according to the difficulty of samples and the unknown degree of domains. This strategy of gradually introducing samples from easy to difficult conforms to the learning law, enabling the model to better adapt to the challenges brought by domain shifts and improving the model's convergence speed and robustness to complex data distributions.

4. In terms of method design, innovative technical architectures and learning strategies are used, as well as theoretical analyses to verify their effectiveness. From the theoretical derivation of the target error to the analysis of the roles of different components, a solid theoretical foundation is provided for the innovation of the method, demonstrating the advantage of the close integration of theory and practice.

**Weaknesses:**

1. In HiLo, features are disentangled by assuming that features from different layers represent domain and semantic information, respectively and minimizing the mutual information based on this assumption. However, this assumption may oversimplify the complexity of feature representation in neural networks. In fact, features from different layers may be a mixture of multiple types of information. Simply defining the early layers as domain features and the late layers as semantic features may not be entirely accurate, which may lead to incomplete feature disentanglement in some complex data distributions and affect the performance and generalization ability of the model.

2. The introduction and extension of PatchMix in the GCD task is an innovation, but it also brings some problems. The adjustment of its objective function and its application on different data increases the complexity of the model. When dealing with data with large domain differences, it is a challenge to determine the mixing proportion and application method accurately. If not handled properly, it may introduce too much noise or incorrect information, which may instead interfere with the learning process of the model and reduce the classification performance.

3. In the curriculum learning method, the adjustment parameters of the sampling probability weights need to be selected through the validation set, which increases the dependence of the model on specific datasets. Moreover, for different datasets and tasks, the optimal values of these parameters may vary greatly, and the model cannot adaptively determine these parameters. If these parameters cannot be correctly selected in a new dataset or task, curriculum learning may not be able to play its intended role. It may even have a negative impact on the learning of the model.

**Questions:**

Please see the Weaknesses.

---

> ### Author Response · Authors · 2024-11-25
>
> > Features are disentangled by assuming that features from different layers represent domain and semantic information….However, this assumption may oversimplify the complexity of feature representation in neural networks.
>
> Our selection of high-level and low-level feature is based on previous literature and empirical evidence. First, several works [S5][S6][S7] have demonstrated the efficacy of separating domain and semantic information and leveraging mutual information to address domain adaptation challenges, showing it is not always that complex when introducing mutual information minimization between the representations. Furthermore, the selection of layers to represent domain and semantic features is not arbitrary but based on extensive empirical investigation. As illustrated in Figure 3 of our paper, we conduct a comprehensive analysis to determine the optimal layer assignments. Our results demonstrate that attaching the domain head to earlier layers yields superior performance, corroborating the hypothesis that lower-level features are more domain-oriented. Likewise, Figure 3(b) shows that fixing the domain head to the first layer and varying the 'Deep' layer for the semantic head from the last to the fourth last layer reveals that the last layer is optimal for the semantic head. These findings substantiate the importance of domain-semantic feature disentanglement and validate our design choice of utilizing lower-level features for domain-specific information and higher-level features for semantic-specific information. In Figure 4(a), we also present the visualization by first applying PCA to the domain features and semantic features obtained through $\mathcal{H}$, and then plotting the corresponding images. As can be seen, the images are naturally clustered according to their domains and semantics, demonstrating that HiLo successfully learns domain-specific and semantic-specific features from higher-level features and the shallower layers.
>
> [S5] Zhao, Haiteng, et al. "Domain adaptation via maximizing surrogate mutual information." IJCAI. 2022.
>
> [S6] Park, Geon Yeong, and Sang Wan Lee. "Information-theoretic regularization for multi-source domain adaptation." ICCV. 2021.
>
> [S7] Sharma, Yash, Sana Syed, and Donald E. Brown. "Mani: Maximizing mutual information for nuclei cross-domain unsupervised segmentation." MICCAI, 2022.
>
> > When dealing with data with large domain differences, it is a challenge to determine the mixing proportion and application method accurately. If not handled properly, it may introduce too much noise or incorrect information, which may instead interfere with the learning process of the model and reduce the classification performance.
>
> We agree that determining appropriate mixing proportions is crucial for PatchMix to be effective. To address this challenge, our method incorporates two dynamic mechanisms for controlling mixing proportions. First, we introduce $\alpha$, which weights each patch based on attention scores from $x$ and $x′$. This ensures that patches less relevant to semantic content receive lower weights in computing $\mathcal{L}^{rep}_s$ and $\mathcal{L}^{cls}_s$, effectively reducing the impact of potentially noisy or irrelevant patches. Second, $\beta$ is sampled from with Beta distribution (Zhu et al. (2023)), as a random mixing proportion for each patch $j$. This dual-control mechanism makes our mixing proportion dynamic rather than static, allowing the model to adaptively adjust the influence of different patches based on their semantic relevance while maintaining a degree of randomness through the Beta distribution. This approach helps mitigate the risk of introducing excessive noise or incorrect information during the mixing process, particularly when dealing with significant domain differences.

---

> > ### Author Response · Authors · 2024-11-25
> >
> > > In the curriculum learning method, the adjustment parameters of the sampling probability weights need to be selected through the validation set, which increases the dependence of the model on specific datasets. Moreover, for different datasets and tasks, the optimal values of these parameters may vary greatly, and the model cannot adaptively determine these parameters. If these parameters cannot be correctly selected in a new dataset or task, curriculum learning may not be able to play its intended role. It may even have a negative impact on the learning of the model.
> >
> > We agree that traditional curriculum learning methods often require careful parameter tuning per dataset, which can indeed increase model dependence on specific datasets. However, our approach fundamentally differs in how we determine sample difficulty. Instead of introducing additional hyperparameters that need validation set tuning, we leverage semi-supervised k-means clustering on domain features extracted from a DINO pretrained backbone to naturally separate samples based on their domain shifts.
> >
> > Specifically, we run semi-supervised k-means on domain features across all domains, where labeled source domain samples serve as anchors. This clustering process automatically identifies samples with varying degrees of domain shift without requiring dataset-specific parameter tuning. Samples that cluster far from source domain centers naturally represent instances with larger domain shifts (higher difficulty), while those clustering closer indicate smaller shifts (lower difficulty). This data-driven approach adapts to the inherent structure of each dataset, making our method more generalizable across different scenarios.
> >
> > While our method does involve some parameters ($r_0$, $r^{\prime}$ and $t^{\prime}$) for controlling the curriculum progression, these parameters are more interpretable compared to traditional curriculum learning parameters. This is because they operate on the natural difficulty hierarchy established by domain shifts rather than arbitrary difficulty measures:
> > - $r_0$ and $r^{\prime}$ represent initial and final sampling ratios for harder samples
> > - $t^{\prime}$ controls the curriculum pacing
> >
> > These parameters follow a simple principle: start with easier samples (closer to source domain) and gradually incorporate harder ones (larger domain shifts). This intuitive progression pattern remains consistent across different datasets, making parameter selection more straightforward and transferable.
> >
> > Moreover, since we use the pretrained DINO backbone for feature extraction, this difficulty assessment is based on general visual representations rather than dataset-specific characteristics. This makes our approach more robust and transferable across different datasets and tasks.

---

### Official Review · Reviewer_6eyX · 2024-11-04

**Soundness:** 3
**Presentation:** 3
**Contribution:** 2
**Rating:** 6
**Confidence:** 4

**Summary:**

This paper introduces a new challenge for Generalized Category Discovery, which requires model to categorize unlabeled data in the presence of domain shifts. Traditional GCD methods assume all images come from the same domain, which leads to a significant performance drop under domain shifts. The proposed HiLo framework explicitly disentangles semantics and domain, achieving domain adaptation in GCD through patchmix and curriculum learning. Experimental results show performance improvements, validating the effectiveness of the approach.

**Strengths:**

1. The paper presents a new, practically meaningful, and challenging setting, and constructs corresponding datasets.
2. The domain-semantic disentangled design is well-reasoned, clearly aligning the motivation.
3. The proposed approach demonstrates significant performance improvement on SSB-C.
4. The writing is clear and easy to follow.

**Weaknesses:**

1. The performance gain on DomainNet is considerably smaller than on SSB-C, and the improvement over methods like SimGCD, which does not account for domain shift, is modest. This indicates limited robustness on various domain shifts and fails to highlight the advantages of the proposed approach.
2. The method is sensitive to certain hyperparameters, and $r^{'}$ does not exhibit consistent performance across the original and new domains.
3. The approach of decoupling domain and semantics is derived from [1], and the use of patchmix for domain adaptation is adapted from [2]. The curriculum learning strategy is also straightforward. Overall, the method seems to be an assembly of prior works, lacking substantial novelty.
4. There is no analysis of the disentangled domain and semantic features, such as distribution visualizations. This would help illustrate the effectiveness of the disentanglement.
5. In line 287, same representation loss $L^{rep}_s$ on both domain and semantic features is confusing. This approach may lead domain features to capture information beyond true domain characteristics. It would be valuable to see t-SNE visualizations of domain features, semantic features, and their combination. The author does not provide a corresponding discussion.
6. Line 313 mentions using pre-trained DINO to obtain $z_d$, but previously $z_d$ is associated with a projection head. If the projection head is discarded, then $z_d$ will always be identical in different time steps. If it is retained, the term “pretrained” is confusing. This needs clarification.
7. The ablation study is somewhat unclear. For instance, in row (5) where only deep features are used, does this mean all other designs related to the shallow feature $z_d$ are also omitted? This also needs clarification.

Reference

[1] Learning deep representations by mutual information estimation and maximization

[2] Patch-mix transformer for unsupervised domain adaptation: A game perspective

**Questions:**

See weakness.

---

> ### Author Response · Authors · 2024-11-25
>
> >  The performance gain on DomainNet is considerably smaller than on SSB-C, which fails to highlight the advantages of the proposed approach.
>
> We appreciate the reviewer's observation regarding the performance differences between SSB-C and DomainNet. We would like to clarify several important points:
>
> First, DomainNet presents a highly challenging scenario due to its dramatic domain shifts and large-scale nature. We find that even ensembling of a number of SoTA domain adaptation methods, combined with SimGCD, show limited improvement on this dataset. For instance, in Table 5, SimGCD+MCC+NWD yields 35.7 ACC, compared to 42.5 for HiLo. Our method achieves 16% improvement in proportional terms for "All" classes ACC on the "Painting" domain, substantially greater than those achieved by combining two SoTA domain adaptation.
>
> Second, the relatively smaller performance gain on DomainNet aligns with a well-known phenomenon in machine learning: improvements on large-scale datasets are typically more modest compared to smaller datasets. For instance, in object detection, SOTA improvements on COCO (large-scale) typically show 1-2% gains [S2], while improvements on smaller datasets like PASCAL VOC can reach 5-7% [S3]. Similarly, in image classification, recent architectural advances show 0.5-1% improvements on ImageNet but 2-3% gains on smaller datasets like CIFAR-100 [S4].
>
> [S2] Sun, P., Zhang, R., Jiang, Y., Kong, T., Xu, C., Zhan, W., ... & Luo, P. “Sparse r-cnn: End-to-end object detection with learnable proposals.” CVPR. 2021.
>
> [S3] Liu, Ze, et al. "Swin transformer: Hierarchical vision transformer using shifted windows." CVPR. 2021.
>
> [S4] Touvron, Hugo, et al. "Training data-efficient image transformers & distillation through attention." ICML, 2021.
>
>
> > The method is sensitive to certain hyperparameters, like $r′$.
>
> As demonstrated in our ablation study (Appendix M), the performance with respect to $r'$ exhibits a convex shape across different configurations, indicating a clear pattern rather than arbitrary sensitivity. Specifically, we found that $r'=0$ performs optimally for the original domain, while $r'=0.5$ works best for corrupted domains. This behavior is actually expected and interpretable: the original domain requires less distribution shifts ($r'=0$) since the data distribution is clean, while corrupted domains benefit from moderate augmentation ($r'=0.5$) to handle distribution shifts.
> Given our challenging setting of simultaneously handling generalized category discovery and domain adaptation, finding a single set of hyperparameters that performs optimally across all domains is inherently difficult. Nevertheless, our method maintains robust performance across domains even with sub-optimal hyperparameter choices (see Appendix M).
>
> > The method seems to be an assembly of prior works, lacking substantial novelty.
>
> Please see the General Response for our response regarding novelty of the method.
>
> > There is no analysis of the disentangled domain and semantic features, such as distribution visualizations
>
> We have visualized disentangled domain and semantic features projected by PCA in Figure 4a, which can effectively show disentanglement. This verifies that HiLo successfully learns domain-specific and semantic-specific features. Additionally, we have added t-SNE visualization to address the next comment below.
>
>
> > Same representation loss Lsrep on both domain and semantic features is confusing… It would be valuable to see t-SNE visualizations of domain features, semantic features, and their combination.
>
> We thank the reviewer for raising this point! Our approach actually addresses a dual GCD problem that operates simultaneously across semantic and domain axes, particularly given that we work without explicit source/target domain splits in unlabeled data. Both semantic and domain spaces contain their own "seen" and "unseen" categories, making representation learning valuable for both aspects to achieve effective disentanglement.
>
> Furthermore, since our PatchMix strategy inherently interweaves both semantic and domain information as part of the augmentation process, maintaining representation learning for both feature types becomes essential for proper disentanglement. We have added t-SNE visualizations in the appendix P that demonstrate that our approach learns distinct domain and semantic features.

---

> > ### Author Response · Authors · 2024-11-25
> >
> > > Clarification of zd in Line 313
> >
> > Indeed, $\mathcal{z}_d$ is computed without the projection head only once before data loading and remains constant across different time steps. This is intentional in our design. For our curriculum sampling, $\mathcal{z}_d$ is used to pre-compute sampling weights before data loading begins. This pre-computation approach is efficient and aligns with our goal of using domain information as a fixed reference point for guiding the learning of semantic features, rather than as a feature representation that needs to be optimized.
> >
> > This design choice helps maintain our focus on clarifying semantic representations while treating domain information as an auxiliary signal for regularization and sampling. We thank the reviewer for raising this point, and we have now further clarified this by removing $\mathcal{z}_d$ in Line 309-310.
> >
> >
> > > In row (5) where only deep features are used, does this mean all other designs related to the shallow feature zd are also omitted?
> >
> > Row (5) means we extract domain features ($\boldsymbol{z}_d$) from the penultimate layer and semantic features ($\boldsymbol{z}_s$) from the final layer. Row (6) means we extract domain features from the first layer and semantic features from the second layer. We have  revised the descriptions to “$\boldsymbol{z}_d, \boldsymbol{z}_s$ from deep features only” and “$\boldsymbol{z}_d, \boldsymbol{z}_s$ from shallow features only” to make it clearer.

---

> > > ### Author Response · Authors · 2024-12-01
> > >
> > > Dear Reviewer 6eyX,
> > >
> > >
> > > Thanks very much for your time and valuable comments. We have provided detailed responses to all your comments and questions point-by-point for the unclear presentations and novelty clarification.
> > >
> > > Any comments and discussions are welcome!

---

> > > > ### Comment · Reviewer_6eyX · 2024-12-01
> > > > **Official Comment by Reviewer 6eyX**
> > > >
> > > > Thank you for the authors' detailed response, which resolves my concerns. As a result, I will increase my score.

---

> ### Author Response · Authors · 2024-12-02
>
> Dear Reviewer 6eyX,
>
>
> We are thrilled to note that your concerns have been addressed. We sincerely appreciate your dedicated time and effort in offering invaluable feedback.

---

### Author Response · Authors · 2024-11-25
**General response**

We thank reviewers for their constructive and valuable feedback. We are encouraged that the reviewers find our paper to be **"clear and easy to follow"** (Reviewer 6eyX), with a **"well-reasoned"** (Reviewer 6eyX) and **“innovative approach”**, while presenting **"innovative technical architectures"** with **"solid theoretical foundation"** (Reviewer RNZQ). The reviewers agreed that our contributions are significant, noting our **"new, practically meaningful, and challenging setting"** (Reviewer 6eyX, jy36) and **"effective feature representation"** learning method (Reviewer RNZQ). We are glad that reviewers also found our evaluation **"demonstrates the model's robustness"** (Reviewer L4eA) and that reviewers acknowledge our **"significant performance improvement on SSB-C"** (Reviewer 6eyX) and effectiveness in handling **"domain shifts"** (Reviewer jy36, L4eA).

We have carefully addressed all concerns raised by the reviewers. First, we provide a **general response** to shared concerns or critical points. We also address the reviewers’ individual concerns after their comments. We have also revised our manuscript based on the comments from the reviewers.


**Novelty of proposed model (Reviewer 6eyX, jy36)**

In this paper we have proposed both a novel (but intuitive) **problem setting** as well as a new **solution** to tackle it. The setting is a challenging and practical image classification problem, as noted by Reviewers 6eyX, RNZQ and jy36. Here, a model must jointly learn from labeled and unlabeled images, with the goal of clustering all unlabeled images into distinct categories. Notably, the unlabeled images may come from different **categories** and **domains** to the labeled set. Though simple and practical, only subsets of this problem have been addressed in prior literature: particularly in the Unsupervised Domain Adaptation (UDA) and Generalized Category Discovery (GCD) fields.

As such, we are transparent in the paper that aspects of our solution have been introduced in prior work (Line 247-248), notably the PatchMix approach. However, we also show that simply ensembling SoTA methods from UDA and GCD does not yield substantial gains in our challenging setting (see Table 5).

Particularly, we find that it is critical to find the optimal recipe for the current task, and that without the following innovations the method does not work:
- A new PatchMix formulation for new classes: PatchMix was developed for UDA and naive application of it does not allow the method to discover new classes. Instead, we must devise a PatchMix-based contrastive learning method to address the challenge of GCD in the presence of domain shift (see Section 3.2.2, and ablation in Table 4). Our approach properly leverages all available samples, including both labelled and unlabelled data, from both in-domain and out-of-domain sources, encompassing both old and new classes. By incorporating these diverse samples, our technique aims to improve the model's ability to handle domain shifts and effectively generalize across different classes.
- A curriculum learning strategy: To our knowledge, the use of curriculums is still underexplored in the GCD literature. We find that in our challenging setting, it is critical for an appropriate curriculum to be introduced.
- Disentangling domain and semantic features: Though the loss formulation has been previously explored, it has not been applied to category discovery before, where it finds a natural fit in the presence of the domain shift problem.

---

### Meta-Review · Area_Chair_yqXe · 2024-12-18

**Metareview:**

The paper introduces HiLo, a novel framework for Generalized Category Discovery (GCD) under domain shifts. It disentangles semantic and domain features using mutual information minimization, enhances learning with PatchMix-based contrastive learning, and integrates curriculum learning. Extensive evaluations on SSB-C and DomainNet benchmarks show substantial improvements over baselines.

The paper's strengths include: 1) Innovative problem setting combining GCD and domain adaptation. 2) Effective disentanglement of domain/semantic features. 3) Robust experimental results with strong theoretical underpinnings.

However, the reviewers raised concerns about the novelty and insufficient analysis of the claimed disentangled feature.

**Additional Comments On Reviewer Discussion:**

During the rebuttal, reviewers raised concerns about novelty, disentanglement assumptions, curriculum learning robustness, and comparison with UniOT. The authors clarified methodological choices, added visualizations, extended comparisons, and refined explanations. These responses effectively resolved most concerns, demonstrating rigorous design and empirical strengths. The paper's practical impact and solid evaluation led to acceptance.

---

### Decision · Program_Chairs · 2025-01-22

Accept (Poster)